# Ionospheric Total Electron Content (TEC) Anomalies as Earthquake Precursors: Unveiling the Geophysical Connection Leading to the 2023 Moroccan 6.8 Mw Earthquake

Karan Nayak [1] , Charbeth López-Urías [1] , Rosendo Romero-Andrade [1,*] , Gopal Sharma [2] , German Michel Guzmán-Acevedo [1] and Manuel Edwiges Trejo-Soto [1]

1 Faculty of Earth and Space Sciences, Autonomous University of Sinaloa, Culiacán 80040, Mexico; nayakkaran.facite@uas.edu.mx (K.N.)

2 North-Eastern Space Application Centre, Umiam 793103, India

* Correspondence: r.romero11@info.uas.edu.mx

**Abstract:** The study delves into the relationship between ionospheric total electron content (TEC) anomalies and seismic activity, with a focus on Morocco's 6.8 Mw earthquake on 8 September 2023, lying within a tectonically active region at the convergence of the African and Eurasian Plates. To enhance the reliability of our findings, we incorporate space weather conditions, utilizing indices (Dst, Kp, and F10.7) to pinpoint periods of stable space weather. This minimizes the possibility of erroneously attributing natural ionospheric fluctuations to seismic events. Notably, our TEC analysis unveils positive and negative anomalies, with some occurring up to a week before the earthquake. These anomalies, exceeding predefined thresholds, provide compelling evidence of significant deviations from typical ionospheric conditions. Spatial mapping techniques employing both station-specific vTEC data and pseudorandom noise codes (PRNs) from multiple global navigation satellite system (GNSS) stations highlight a strong correlation between ionospheric anomalies and the earthquake's epicenter. The integration of PRNs enhances coverage and sensitivity to subtle anomalies. Additionally, the analysis of satellite imagery and ground displacement data using Sentinel-1 confirms significant ground uplift of approximately 15 cm following the earthquake, shedding light on surface responses to seismic events. These findings underscore the potential of ionospheric science in advancing earthquake early warning systems and deepening our understanding of earthquake precursors, thus contributing to the mitigation of seismic event impacts and the protection of lives and infrastructure.

**Keywords:** Morocco earthquake; total electron content; ionospheric anomalies; PRNs; InSAR; LAIC

## 1. Introduction

The study of seismic activity and its precursors has long been a subject of profound scientific interest and societal importance. Earthquakes, being natural disasters of considerable magnitude, have the potential to cause devastating consequences to both human life and infrastructure. Understanding the intricate processes leading to an earthquake is crucial for early warning systems and disaster preparedness. Traditional earthquake prediction methods primarily rely on monitoring geological and seismological parameters, such as ground deformation [1–3], strain accumulation [4,5], lineament analysis [6,7], and seismic activity [8]. However, recent advances in the field of ionospheric science have opened new avenues for earthquake forecasting, offering complementary insights into the complex processes that precede seismic events. One emerging area of research that holds promise in unraveling the mysteries of earthquake precursors is the analysis of ionospheric total electron content (TEC) anomalies [9–15]. This study explores the intriguing relationship between ionospheric TEC and seismic activity, focusing specifically on the North African region, with a primary emphasis on Morocco. Situated in a tectonically active zone

at the convergence of the African and Eurasian Plates, Morocco has experienced notable seismic events throughout its history [16]. Understanding the ionospheric TEC response as a potential earthquake precursor in this region holds significant promise for advancing earthquake early warning systems and enhancing our understanding of the underlying geophysical processes.

The ionosphere, a region of Earth's upper atmosphere, plays a pivotal role in the propagation of radio waves and the global navigation satellite system (GNSS). Perturbations in ionospheric TEC have been observed before and after earthquakes, offering a tantalizing possibility of detecting precursory signals that may precede seismic events [17–19]. However, the mechanisms governing the ionospheric response to tectonic stress and the temporal relationship between TEC anomalies and earthquakes remain subjects of active research. TEC is a fundamental parameter in ionospheric science that characterizes the total number of electrons present along a specific path through the Earth's ionosphere [20]. The ionosphere, a region of the Earth's upper atmosphere, is a complex, dynamic layer consisting of charged particles, primarily electrons and ions, that are influenced by various solar and terrestrial processes. TEC is a critical measure of the ionosphere's electron density and is typically expressed in units of electrons per square meter ($el/m^2$) along a given path [21]. TEC values exhibit considerable variability influenced by factors such as location, time of day, and solar activity. These values respond to solar radiation, geomagnetic activity, and interactions with Earth's magnetic field. Key aspects of TEC encompass its diurnal variation, peaking around local noon due to maximum solar ionization and decreasing as the sun sets [22]. Solar activity, including solar flares and sunspots, significantly affects TEC levels by enhancing ionization in the ionosphere [23]. Geomagnetic activity resulting from solar interactions with Earth's magnetic field leads to TEC variations [24], while events like seismic activity and atmospheric disturbances can also influence TEC measurements, making it relevant in earthquake precursor research [12–15,17–19,25]. TEC monitoring primarily utilizes signals from the GNSS, including GPS satellites. As GNSS signals traverse the ionosphere, they experience delays and frequency shifts due to electron density fluctuations, enabling the quantification and mapping of TEC across specific geographic regions [26]. TEC's applications extend beyond earthquake precursors, serving as a crucial component in fields like satellite-based navigation and communication systems, where accurate measurements are vital for mitigating ionospheric effects on GNSS signals, including signal degradation and positioning errors [27]. Additionally, TEC measurements have found application in space weather monitoring, including the detection and analysis of solar flares, where rapid variations in TEC can serve as valuable indicators of solar-induced ionospheric disturbances [28].

Within earthquake research, TEC has emerged as a potential tool for monitoring ionospheric anomalies preceding seismic events, contributing to our understanding of earthquake precursors, and enhancing early warning systems. In this ongoing investigation, we embark on a thorough analysis of TEC data leading up to the Morocco earthquake. Our objective is to elucidate TEC's role in detecting seismic activity while validating its findings with ground-based observations.

## 2. Study Area and Seismotectonic Setting

Our study focuses on the region encompassing the Morocco earthquake of 6.8 magnitude (Mw), which occurred on 8 September 2023 with a depth of 19 km (https://earthquake.usgs.gov/, accessed on 20 September 2023). The epicenter of this seismic event was located at 31.055° N 8.389° W, resulting in significant impacts on the surrounding areas. The investigation extends to the immediate vicinity of the epicenter to analyze the seismic and ionospheric phenomena leading up to this event. Morocco's seismotectonics are characterized by its location at the convergence of the African Plate and the Eurasian Plate, making it prone to seismic activity. This geological setting gives rise to diverse features such as mountain ranges, fault systems, and a history of seismic events. The North Alboran Fault, extending offshore into the Alboran Sea, plays a crucial role in the seismicity of the region,

marking the boundary between the African and Eurasian Plates [29]. It is closely monitored due to its potential for significant earthquakes. The Alboran Sea area is recognized for its history of seismic activity, impacting both Morocco and neighboring regions. Furthermore, Morocco is intersected by multiple fault systems, including those associated with the Rif Mountain range and the Atlas Fault System [30]. These fault systems contribute to the accumulation and release of seismic stress, rendering Morocco susceptible to earthquakes. The earthquake of 6.8 magnitude (Mw) under investigation represents a significant event within Morocco's seismotectonic context. Understanding the geological and tectonic factors leading to this earthquake is vital for assessing its seismic hazard and identifying potential ionospheric precursors.

In addition to Morocco, the broader Mediterranean region experiences heightened seismic activity due to the northward convergence (4–10 mm/year) of the African plate towards the Eurasian plate along a complex plate boundary [31,32]. This process began approximately 50 million years ago during the closure of the Tethys Sea, with the Mediterranean Sea remaining as its modern-day remnant. The region's most substantial seismicity rates are observed along the Hellenic subduction zone in southern Greece, the North Anatolian Fault Zone in western Turkey, and the Calabrian subduction zone in southern Italy [29]. These regions have varying rates of tectonic convergence, leading to different types of faulting and seismic activity. Notable instances include the 2023 Turkey earthquake sequence, marked by a Mw 7.8 earthquake near the Syrian border, followed by a Mw 7.5 quake 90 km to the north, both along the East Anatolian Fault Zone—a left-lateral strike-slip fault separating the Anatolian Plate from the Arabian Plate [33,34]. These studies, using diverse methods like image fusion and seismic source modeling [33–35], reveal the complex geodynamic setting in the broader region, characterized by a rich seismic history and numerous active faults with diverse directions and kinematics. Furthermore, the Mediterranean region boasts a centuries-long written record documenting pre-instrumental seismicity (prior to the 20th century). Earthquakes have historically wrought widespread destruction in several Mediterranean regions, and tsunamis triggered by large earthquakes have had significant impacts. The 1755 Lisbon earthquake is a historical example, with an estimated magnitude of approximately 8.0. This earthquake is believed to have occurred within or near the Azores-Gibraltar transform fault, marking the boundary between the African and Eurasian plates off the western coasts of Morocco and Portugal. It is noteworthy not only for its high casualty count, approximately 60,000 lives lost, but also for generating a tsunami that surged along the Portuguese coast, inundating coastal settlements and Lisbon [36]. Similarly, in 1693, an earthquake of around M7.4 near Sicily produced a substantial tsunami wave that devastated numerous towns along Sicily's eastern coastline [37]. Europe's deadliest documented earthquake, the M7.2 28 December 1908 Messina earthquake, resulted in an estimated 60,000 to 120,000 fatalities due to the combination of intense ground shaking and a local tsunami [38].

The studied earthquake resulted from shallow oblique-reverse faulting in the High Atlas Mountains, 75 km southwest of Marrakech. It occurred within the Africa Plate, about 550 km south of the Africa-Eurasia plate boundary, where the African Plate moves about 3.6 mm/year west-southwest relative to the Eurasia Plate [39]. Following the main quake, four aftershocks were recorded with magnitudes ranging from 4 to 5 Mw. The most significant aftershock, measuring 4.9 Mw, happened just 20 min after the main shock. Earthquakes of this magnitude in the area are infrequent but within the realm of possibility. Since 1900, there have been nine earthquakes with a magnitude of 5 or greater within a 500 km radius of this event. However, none of these previous earthquakes exceeded a magnitude of 6. Most of these past events have occurred to the east of the earthquake that took place on 8 September 2023.

## 3. Data Used and Methodology

### 3.1. Data Collection and Pre-Processing

To investigate the relationship between ionospheric TEC variations and seismic activity leading up to the Morocco earthquake, we compiled a comprehensive dataset of GNSS signals from 11 stations (https://cddis.nasa.gov/, accessed on 9 September 2023). GNSS, including GPS, provides an invaluable source of data for TEC calculation due to its widespread availability and continuous transmission of signals from satellites in orbit. We accessed GNSS data from ground-based stations strategically located in Morocco and the surrounding regions (Figure 1). These stations were selected to ensure comprehensive spatial coverage, enabling us to monitor TEC variations over a broad area. Also, the stations lie within the earthquake preparation zone, which is estimated to be 840 km, as per the Dobrovolsky equation [40].

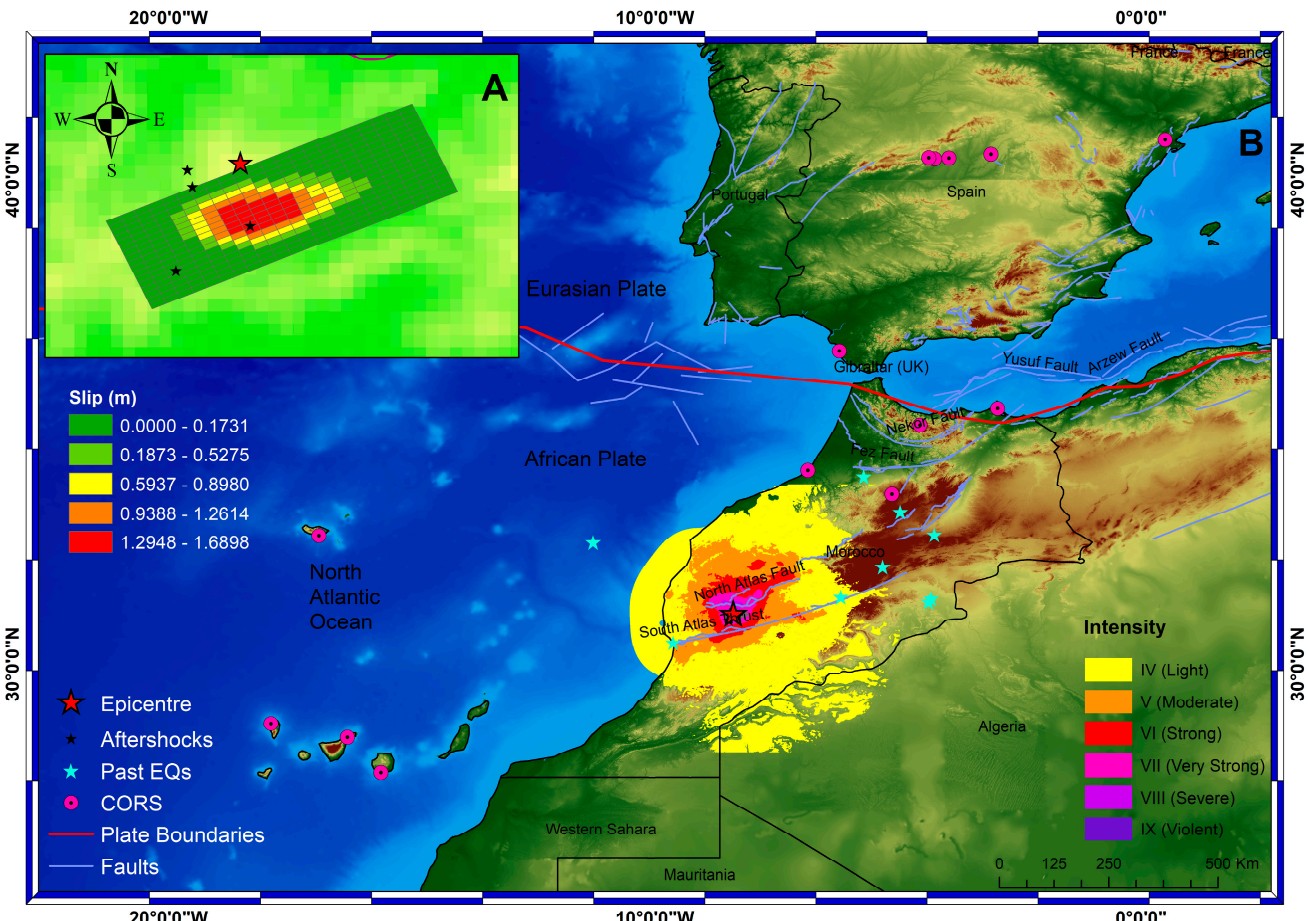

**Figure 1.** The seismotectonic of the study region highlighting the epicenter of the Mw 6.8 earthquake with the red star. (**A**) Incorporating finite faults in the analysis for a comprehensive view of seismic activity while superimposing the slip distribution on GEBCO bathymetry. Black stars are aftershock locations. (**B**) The map indicates earthquake intensity, with the Morocco earthquake marked by a red star. Cyan stars denote previous seismic events in the region with a magnitude exceeding 5. Violet lines represent fault lines, while the topographic map source is CGIAR SRTM (30 s) in Grid Format.

The GNSS signal data underwent rigorous preprocessing to ensure accuracy and consistency. This included correcting for various sources of signal delay, such as tropospheric effects and satellite clock errors. We also applied quality control measures to remove outliers and noise from the dataset. To check the influence of Solar flares and geomagnetic storms, we acquired the data from NOAA (https://www.noaa.gov/, accessed on

9 September 2023). The details of the stations used for the analysis are given below in Table 1.

**Table 1.** Details of the GNSS stations used for the study.

| ID | Lat | Long | Array | ID | Lat | Long | Array |
|------|--------|---------|-------|------|--------|---------|-------|
| RABT | 33.998 | −6.854 | IGS | MADR | 40.429 | −4.25 | IGS |
| SFER | 36.464 | −6.206 | IGS | CEBR | 40.453 | −4.368 | IGS |
| EBRE | 40.821 | 0.492 | IGS | FUNC | 32.648 | −16.90 | IGS |
| LLAG | 28.482 | −16.32 | IGS | LPAL | 28.764 | −17.89 | IGS |
| YEBE | 40.525 | −3.089 | IGS | MAS1 | 27.764 | −15.63 | IGS |
| VILL | 40.444 | −3.952 | IGS | | | | |

*3.2. TEC Calculation*

TEC calculation is central to our investigation, as it provides a measure of the electron density in the ionosphere along the signal paths of the GNSS satellites. The Faraday rotation and time delay of GNSS signals as they pass through the ionosphere offer insights into TEC variations. We employed dual-frequency GNSS receivers to compute TEC values. This method involves measuring the difference in phase between the $L_1$ (1575.42 MHz) and $L_2$ (1227.60 MHz) carrier frequencies of GNSS signals. The sTEC value (slant TEC) along each slant trajectory satellite-receiver path can be estimated using the following Equation (1) [41]:

$$sTEC = \frac{f_1^2 \cdot f_2^2}{40.3082 \left(f_1^2 - f_2^2\right)} \left\{ (P_2 - P_1) - \left(b_p^s + b_p^r\right) \right\} \tag{1}$$

where the carrier frequencies of the GPS signals are denoted by $f_1$ and $f_2$ ($f_1$ = 1575.42 MHz and $f_2$ = 1227.6 MHz), $P_1$ and $P_2$ are the pseudo ranges corresponding to the carrier frequencies, $b_p^s$ is the satellite bias, and $b_p^r$ is the receiver bias.

sTEC values vary with the satellite's elevation angle, so it is essential to calculate the vTEC (vertical TEC). vTEC is determined using a mapping function (Equation (2)), which assumes the ionospheric point of interest is at a height of 350 km [42].

$$vTEC = sTEC \times cos \left[ arcsin \frac{R cos \left(\theta\right)}{R + h} \right] \tag{2}$$

where R represents the Earth's radius (6371.2 km), h is the ionosphere's height above the Earth's surface (350 km), and $\theta$ denotes the elevation angle, which is measured in degrees between the satellite and the receiver.

A robust statistical foundation for data analysis, employing a 15-day moving average and standard deviation, is used to establish upper and lower boundaries as illustrated in Equation (3)

$$Boundaries = X \pm 1.34\sigma \tag{3}$$

where *X* represents the mean and σ signifies the standard deviation. By calculating these boundaries, this approach ensures that the limits are statistically grounded and responsive to the data's inherent variability. The selection of 1.34σ as the boundary multiplier in Equation (3) is a critical decision in our methodology. We opted for 1.34σ to balance sensitivity and specificity in anomaly detection. This choice, corresponding to a 90.99% probability level, ensures that the anomalies identified are statistically significant while minimizing the risk of false positives, aligning with our research's goal of detecting genuine variations in TEC patterns. Importantly, our decision to adopt a Gaussian distribution is grounded in prior research [13,19]. The absence of a specific confidence level calculation can be attributed to our reliance on the Gaussian distribution assumption. Any instances

where the data values exceed these boundaries are classified as anomalies, denoting points that significantly deviate from the anticipated norms. Furthermore, Equations (4) and (5) are used to pinpoint the peak anomaly times for both peak positive (*PPA*) and peak negative (*PNA*) anomalies, providing valuable insights into when and how these deviations occur within the dataset, enriching the understanding of the underlying patterns and trends.

$$PPA = vTEC - UB \tag{4}$$

$$PNA = LB - vTEC \tag{5}$$

where UB and LB are upper and lower boundaries, respectively. vTEC is the vertical TEC.

### 3.3. Spatial Mapping

vTEC values were spatially mapped across the study region, providing a visual representation of electron density variations. This mapping was essential for identifying anomalous TEC patterns and tracking their evolution leading up to the Morocco earthquake. Accurate spatial representation of vTEC is essential for identifying anomalous patterns and their association with seismic activity. Kriging is a geostatistical technique that plays a pivotal role in generating spatial maps of vTEC. This interpolation method is particularly well-suited for our research because it considers not only the sampled data points but also the spatial autocorrelation and variability of the vTEC field. Kriging interpolation provides a robust means to estimate vTEC values at unobserved locations, creating a continuous spatial representation while minimizing the estimated variance [43]. Using the selected semivariogram model, we perform Kriging estimation to predict vTEC values at unobserved locations within the study area. Kriging takes into account the spatial relationship between observed data points and assigns weights to neighboring points based on their proximity and spatial autocorrelation [44]. The resulting vTEC spatial map provides a continuous representation of electron density variations across the region of interest. This map can reveal spatial gradients, anomalies, and trends in vTEC that may be associated with geological and ionospheric phenomena, including those potentially linked to seismic activity. This integrated approach, combining spatial mapping and seismic data analysis, enhances our ability to assess the utility of vTEC as a potential seismic precursor.

## 4. Results and Discussions

### 4.1. Space Weather Analysis

The ionosphere can experience disturbances due to various factors, which include geomagnetic storms, solar activity, atmospheric waves, and changes in lower atmospheric pressure. It is important to note that solar activity and geomagnetic storms have the potential to generate disruptions in the ionosphere that might be wrongly interpreted as anomalies related to earthquakes if not appropriately accounted for. The ionosphere's structure also varies significantly with latitude, including the equatorial anomaly with electron concentration peaks formed in the afternoon due to an eastward electric field near the geomagnetic equator. Therefore, it is essential to rule out these factors and ensure that the observed irregularities are genuinely connected to seismic events. In this study, three specific indicators, namely Dst, Kp, and F10.7, were utilized to evaluate the calmness of space weather conditions. Elevated levels of solar or geomagnetic activity can trigger deviations in the ionosphere, especially during periods of heightened activity. The Dst (disturbance storm time) index is a measure of geomagnetic activity. It quantifies disturbances in the Earth's magnetic field resulting from interactions with the solar wind. When these disturbances are significant, Dst values become negative. In our analysis, negative Dst values indicate geomagnetic storms. To ensure that ionospheric anomalies are not attributed to these storms, we set a threshold of Dst > −50 nT. This threshold assures that the Earth's magnetic field remains relatively undisturbed during the observed period. However, it should be noted that the Dst value reached −60 nT on 2 September 2023, which may suggest a moderate to relatively mild geomagnetic storm. Dst values less than −100 nT

signify a severe or major geomagnetic storm (NOAA). The Kp index provides a global perspective on geomagnetic activity. It is rated on a scale from 0 to 9, with higher values denoting increased geomagnetic activity. We consider periods with Kp values below 5 as relatively calm in terms of geomagnetic conditions. This threshold is crucial for confirming that the ionospheric disturbances are not caused by heightened geomagnetic activity. The F10.7 (solar radio flux) index relates to solar activity, specifically emissions from the Sun at a frequency of 10.7 cm. A value below 150 indicates low solar activity, suggesting fewer solar emissions affecting the ionosphere. By setting a threshold of F10.7 < 150, we ensure that our analysis focuses on times when solar activity is at an average to low level. This allows us to rule out solar-related influences on ionospheric anomalies.

Figure 2 illustrates the fluctuations in each index throughout the entirety of August 2023 up to the date of the earthquake, which occurred on 8 September 2023. The findings indicate that solar activity exhibited strength from 26 July to 16 August. Additionally, significant geomagnetic disruptions were observed on August 4th and September 2nd and 3rd, with the Kp index surpassing 5 and the Dst index falling below −60 nT. In light of these results, when assessing the unusual ionospheric disturbances detected during the mentioned periods, it is advisable to initially consider the contributory factors of solar and geomagnetic activity. Figure 3 further highlights the detailed depiction of space weather conditions, offering a comprehensive overview of ionospheric variations during the period spanning 31 August to 3 September 2023. This figure presents data points collected at three-hour intervals, allowing for a more finely grained analysis of the observed space weather patterns leading up to the 2023 Moroccan earthquake.

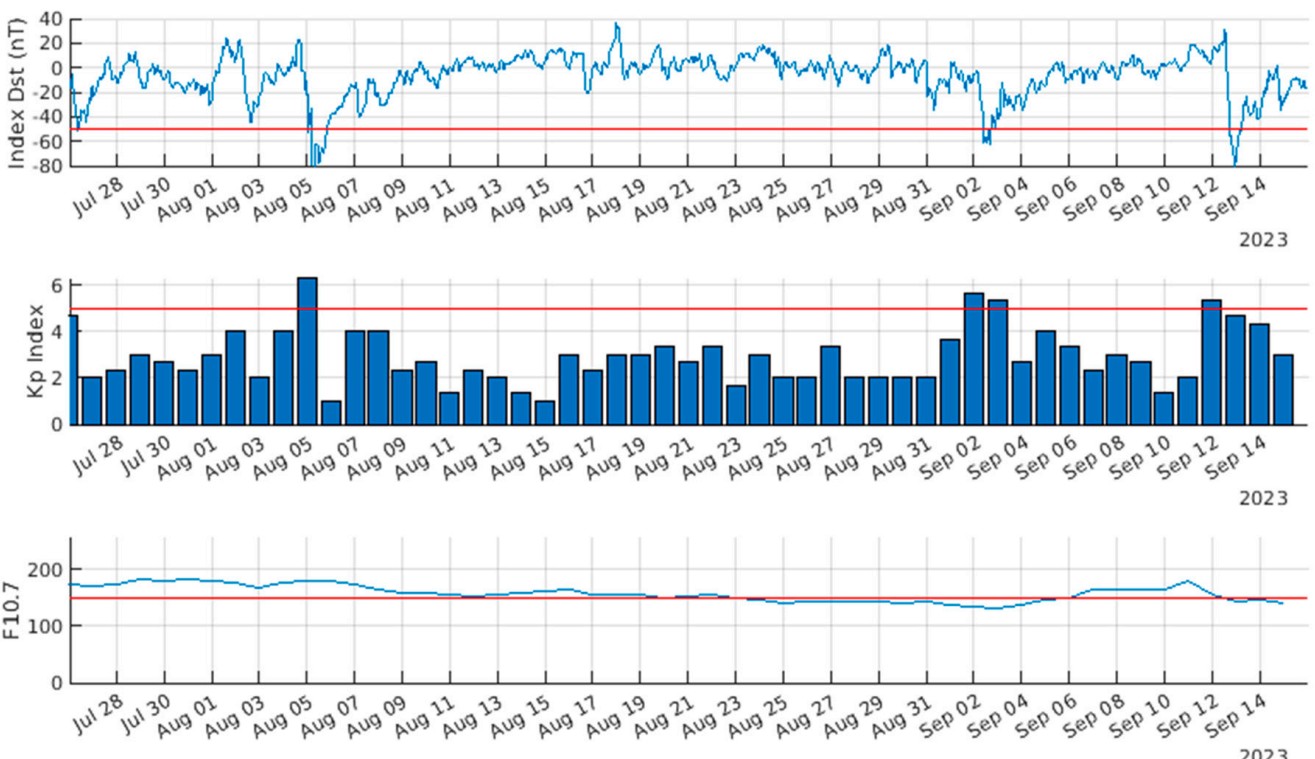

**Figure 2.** This figure provides a detailed representation of the daily variations in three significant space weather indices, specifically the Dst index, the Kp, and the F10.7 index. These variations are observed during the time frame spanning from 26 July to 13 September, encompassing the 45 days leading up to the earthquake and extending one week beyond it. The red horizontal lines, evident in each sub-plot, indicate the corresponding threshold levels established for these indices.

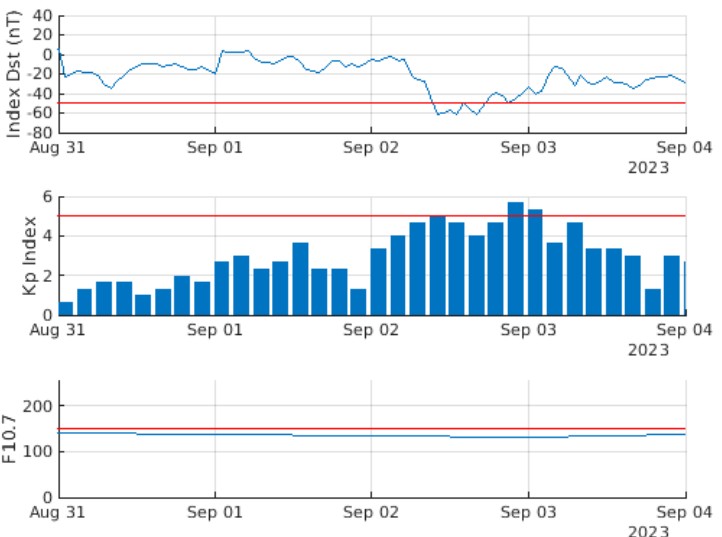

**Figure 3.** Detailed space weather observations—31 August to 3 September 2023. This figure presents a granular view of space weather conditions, featuring data points captured at three-hour intervals during the specified period, providing a more comprehensive perspective of ionospheric variations in the lead-up to the 2023 Moroccan earthquake.

### 4.2. TEC Analysis

The earthquake preparation zone, spanning an impressive 840 km, was meticulously calculated. This region encompasses 11 IGS CORS (international GNSS service continuously operating reference station) stations, a visual representation of which can be observed in Figure 1. For a closer examination of TEC variations leading up to the earthquake event, we leveraged data collected from the nearest CORS station, RABT, situated approximately 355 km north of the epicenter. Figure 4 vividly illustrates the TEC fluctuations within the 30-day window preceding the earthquake event, which transpired on 8 September 2023. Our TEC analysis yielded a compelling set of results. Notably, during the period of 17–19 August and 22 August 2023, as well as 2–3 September 2023, a series of conspicuous positive anomalies were identified. Moreover, a significant positive anomaly was detected on the precise day of the earthquake, 8 September 2023. Negative anomalies were also observed to be a common occurrence, amplifying their significance on both 1 September and 3 September. In Figure 4, the black and green columns correspond to instances where TEC values exceeded the upper and lower thresholds, respectively, defined in TECU (total electron content units). Particularly noteworthy was the positive anomaly on 2 September 2023, occurring six days before the earthquake, with TEC concentrations surpassing 10 TECU. Conversely, a noteworthy negative anomaly manifested on 1 September 2023, with TEC values falling below 5 TECU, a full seven days prior to the main seismic event.

The findings extracted from the closest station to the epicenter, RABT, provide compelling evidence of significant anomalies closely associated with the 6.8 Mw earthquake. In our quest to pinpoint the exact timing of peak anomalies, we employed Equations (4) and (5). The PPA was determined to occur on 2 September 2023 at 13.7 UTC, exhibiting a substantial difference of 10.449 TECU. In contrast, the PNA was calculated to transpire on 1 September 2023 at 18.75 UTC, characterized by a difference of 5.163 TECU. It is crucial to emphasize that these PPA and PNA values were established while considering not only space weather phenomena but also the latitudinal variations, with an emphasis on the afternoon as the period of highest significance. These anomalies were observed within a tranquil environmental context, bolstering the case for their potential relevance as seismic precursors.

In our endeavor to comprehensively analyze the spatial and temporal patterns of vTEC, we pursued two distinct approaches. The initial approach centered on a meticulous analysis of data from the 11 designated stations located within the earthquake preparation zone. This involved calculating vTEC values during the specific timeframes corresponding

to the peak anomalies, encompassing both positive and negative deviations from the norm. These focused calculations allowed us to scrutinize the behavior of vTEC within this critical window, providing insights into the ionospheric response preceding seismic events. Conversely, our second approach broadened the scope of our investigation by encompassing PRNs (pseudorandom noise codes) from all available GNSS stations. This approach, too, involved examining vTEC variations during the same specified peak anomaly times. By including PRNs from multiple sources, we aimed to achieve a more expansive and comprehensive understanding of the vTEC patterns during these crucial periods.

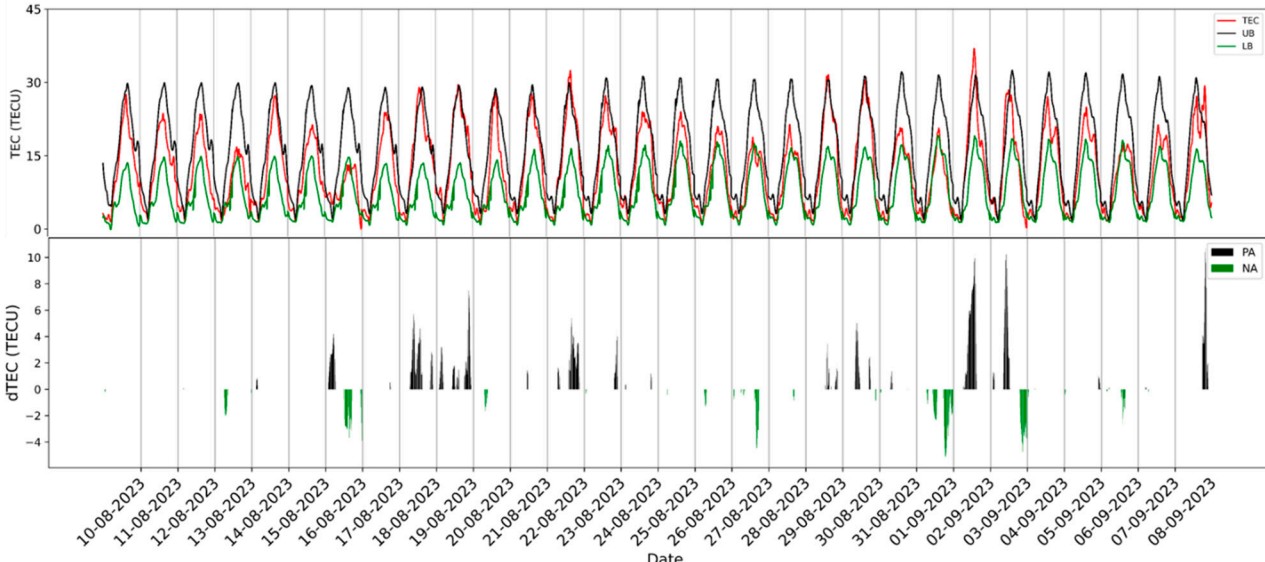

**Figure 4.** Daily TEC measurements, expressed in TECU, were taken at the nearest RABT station during the 30 days leading up to the earthquake. The upper and lower limits, defined by Equation (3), are depicted by solid black and green lines, respectively. The red solid line illustrates the daily variations in TEC. Any deviations from these limits, indicated by the black and green columns, were considered anomalies. The graph below shows the TEC variance relative to the lower boundary, measured in TECU.

### 4.2.1. Positive Anomaly

The positive anomaly was detected on 2 September, approximately six days preceding the seismic event. The calculated PPA was at 13.7 UTC, having a difference of more than 10 TECU. Incorporating the timing of the PPA, we proceeded to create spatial representations of vTEC using our dual methodologies: one based on station-specific vTEC and the other based on vTEC derived from PRNs precisely at the identified PPA. Our dataset was meticulously assembled from TEC measurements obtained from an array of 11 IGS stations. The outcomes of our spatial mapping efforts, conducted with station-specific vTEC data, are vividly illustrated in Figure 5. A noteworthy observation from this depiction is the close alignment of the anomaly zone with the epicenter of the impending earthquake. Notably, the RABT station, being closest to the epicenter, exhibited the most heightened TEC concentration among all stations, signifying its pivotal role in capturing ionospheric anomalies associated with the impending seismic event.

Figure 5A offers an insightful visualization of the earthquake epicenter, inclusive of finite fault lines and their respective slip rates. This visualization highlights a correlation between the region surrounding the earthquake epicenter, its associated active faults, and an increase in TEC values, aligning with the positive anomaly observed at the PPA. Figure 5B further contributes to our understanding by presenting a spatiotemporal distribution of GPS TEC anomalies, specifically centered on the identified PPA time of 13.7 UTC. Within this framework, the most prominent positive TEC anomaly was pinpointed to the northeast of the epicenter. Remarkably, the RABT station, with its proximity to this region,

emerged as the station most significantly impacted by these ionospheric anomalies, further underscoring its importance in the context of seismic events. These findings collectively underscore the intricate interplay between ionospheric anomalies and seismic events within our study area, reinforcing the value of our comprehensive approach.

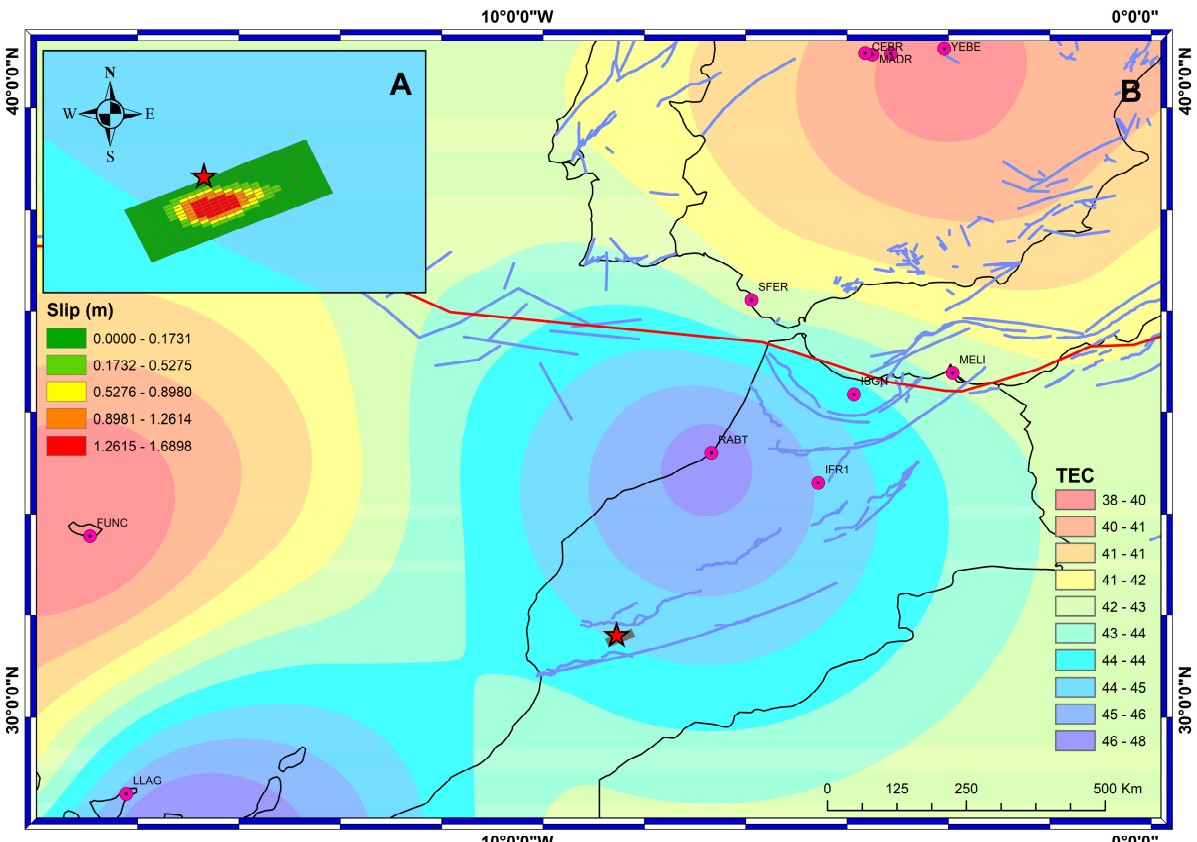

**Figure 5.** Spatial distribution of vTEC as per station, which are illustrated by pink circles for the peak positive anomaly time at 13.7 UTC. (**A**) represents the TEC behavior within the earthquake's epicenter region and the related finite faults. (**B**) represents the TEC behaviour within the seismopreparation zone corresponding to the monitoring stations.

For our second spatial mapping approach, we opted to analyze vTEC using PRNs specific to each ground-based station. The outcomes of this approach are illustrated in Figure 6, showcasing the spatial distribution of vTEC in relation to the available PRNs at the PPA occurring at 13.7 UTC.

In Figure 6A, a visual representation of the earthquake epicenter, with finite fault lines and their respective slip rates, is presented. It is evident that the highest concentration of TEC values was observed in close proximity to the epicenter. This visual representation underscores a compelling correlation between the geographic area surrounding the earthquake epicenter, the activity of its associated fault lines, and an elevation in TEC values. This alignment with the positive anomaly observed at the PPA underscores the relevance of ionospheric variations in the lead-up to seismic events. Figure 6B further enriches our understanding by offering a spatiotemporal perspective of GPS TEC anomalies, taking into account the available PRNs at the identified PPA time of 13.7 UTC. Intriguingly, the anomaly was consistently situated within the epicentral region, irrespective of the station's geographic location. This approach yields promising results in terms of approximating the epicenter's location based on the anomaly and demonstrates a strong correlation between fault lines and the distribution of TEC value, indicating that PRNs capture ionospheric variations more holistically. This approach not only excels in approximating the epicenter's location based on the anomaly but also provides a broader spatial context for ionospheric

anomalies, which is crucial for understanding their relationship with seismic activity. These collective findings emphasize the intricate interplay between ionospheric anomalies and seismic events within our study area.

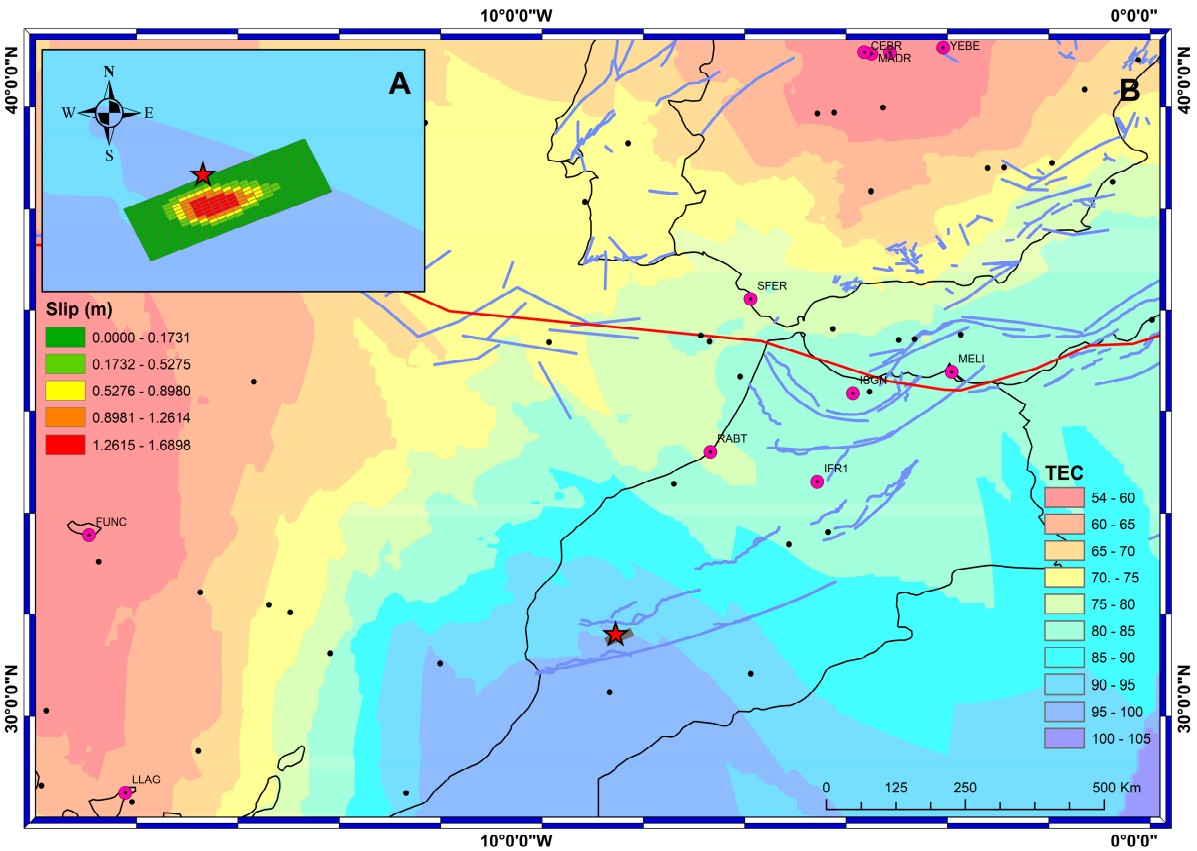

**Figure 6.** Spatial distribution of vTEC as per satellite PRNs, which are illustrated by black circles for the peak positive anomaly time at 13.7 UTC. (**A**) represents the TEC behaviour within the earthquake's epicenter region and the related finite faults. (**B**) represents the TEC behaviour within the seismopreparation zone corresponding to the satellite PRNs.

### 4.2.2. Negative Anomaly

The presence of a negative anomaly was discerned on 1st September, with the peak negative anomaly (PNA) pinpointed at 18.75 UTC, demonstrating a deviation of approximately 5 TECU from the established norm. Employing the same two spatial mapping approaches used for positive anomalies, we sought to understand the spatial distribution of vTEC during this period.

In the first method, which involved mapping vTEC with respect to individual ground stations, as showcased in Figure 7, we gained insights into the behavior of the ionosphere at the time of the peak negative anomaly. Figure 7A provides a comprehensive view of the earthquake epicenter and the associated finite fault lines. This visual representation revealed a noticeable impact on the ionosphere within the epicentral region, characterized by a low TEC zone. Figure 7B takes our analysis a step further by presenting a spatiotemporal distribution of GPS TEC anomalies, considering data from all 11 available stations. Notably, this analysis revealed that the epicentral zone was situated within a region of low TEC. Additionally, the nearest station, RABT, exhibited comparably lower TEC values. As we extended our analysis from RABT to other nearby stations, we observed a discernible decrease in TEC values. This decline in TEC along this trajectory suggests a possible association between the detected anomaly and seismogenic activity.

The second vTEC mapping approach utilizing PRNs is depicted in Figure 8, providing additional insights into the spatial distribution of vTEC during negative anomalies.

Figure 8A presents a concentration of TEC values, highlighting the epicentral region and the faults associated with the earthquake. Notably, TEC concentration was consistently lower in this region, indicating a robust correlation between active faults, the earthquake epicenter, and ionospheric TEC levels. This alignment underscores the influence of seismic activity on ionospheric behavior. Figure 8B augments our understanding by presenting a spatiotemporal view of GPS TEC anomalies, accounting for the available PRNs at the identified PNA time. Intriguingly, the negative anomaly was consistently centered within the epicentral region, regardless of the geographic location of the ground-based stations, mirroring the pattern observed for positive anomalies.

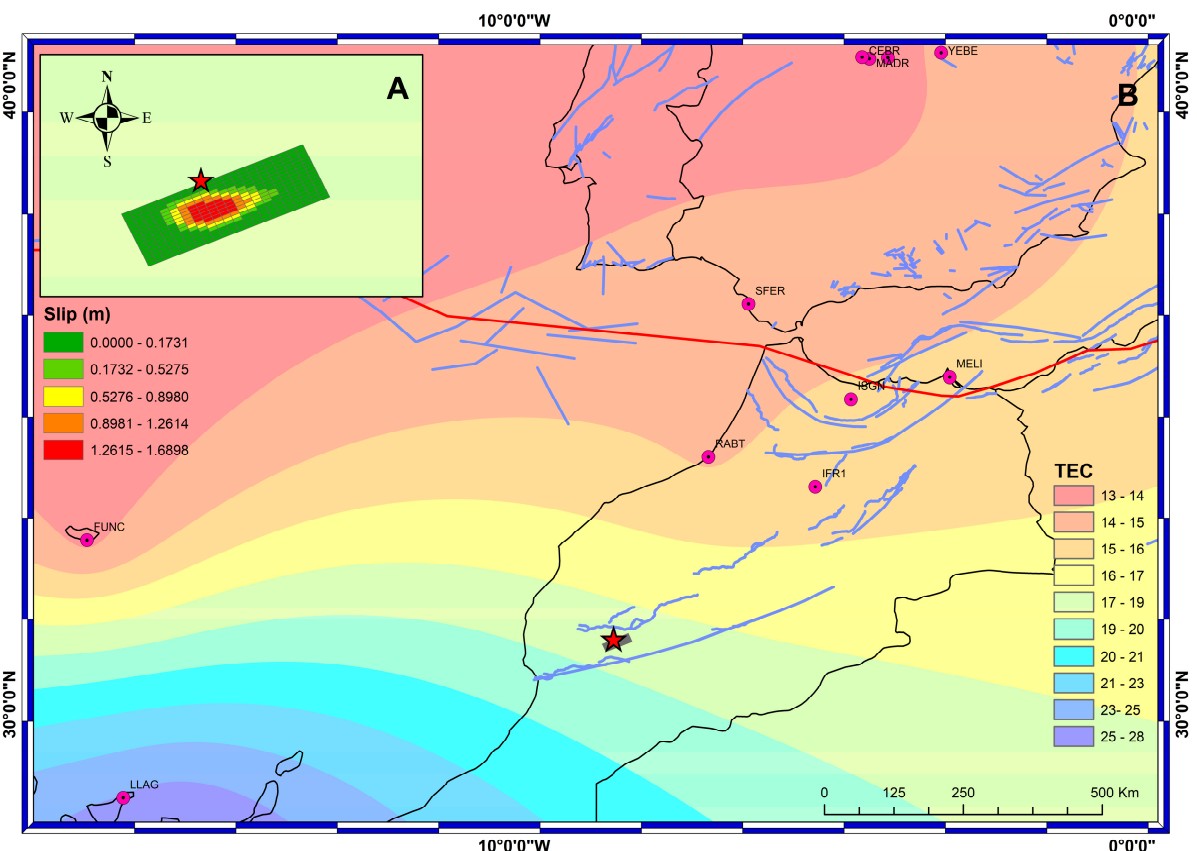

**Figure 7.** Spatial distribution of vTEC as per station, which are illustrated by pink circles for the peak negative anomaly time at 18.75 UTC. (**A**) represents the TEC behaviour within the earthquake's epicenter region and the related finite faults. (**B**) represents the TEC behaviour within the seismo-preparation zone corresponding to the monitoring stations.

Based on the aforementioned results, the decision to integrate PRNs into our spatial mapping methodology, in contrast to exclusively relying on traditional station-based vTEC data, holds considerable significance for multiple compelling reasons. PRNs furnish a more extensive and finely tuned perspective of the Earth's ionosphere, as they are associated with a multitude of GNSS satellites. This broader coverage not only allows us to capture a more comprehensive array of spatial and temporal vTEC variations but also provides a richer understanding of ionospheric behavior leading up to seismic events. Furthermore, the utilization of PRNs significantly heightens our capacity to identify subtle anomalies and deviations, which may not be as apparent when exclusively depending on a limited number of fixed ground stations. Consequently, the integration of PRNs into our spatial mapping approach proves indispensable for a more robust and enlightening analysis of vTEC dynamics concerning seismic events. Importantly, this method's independence from station location facilitates the approximation of anomalies in proximity to the epicentral region, ensuring a more comprehensive assessment of ionospheric responses to seismic activity.

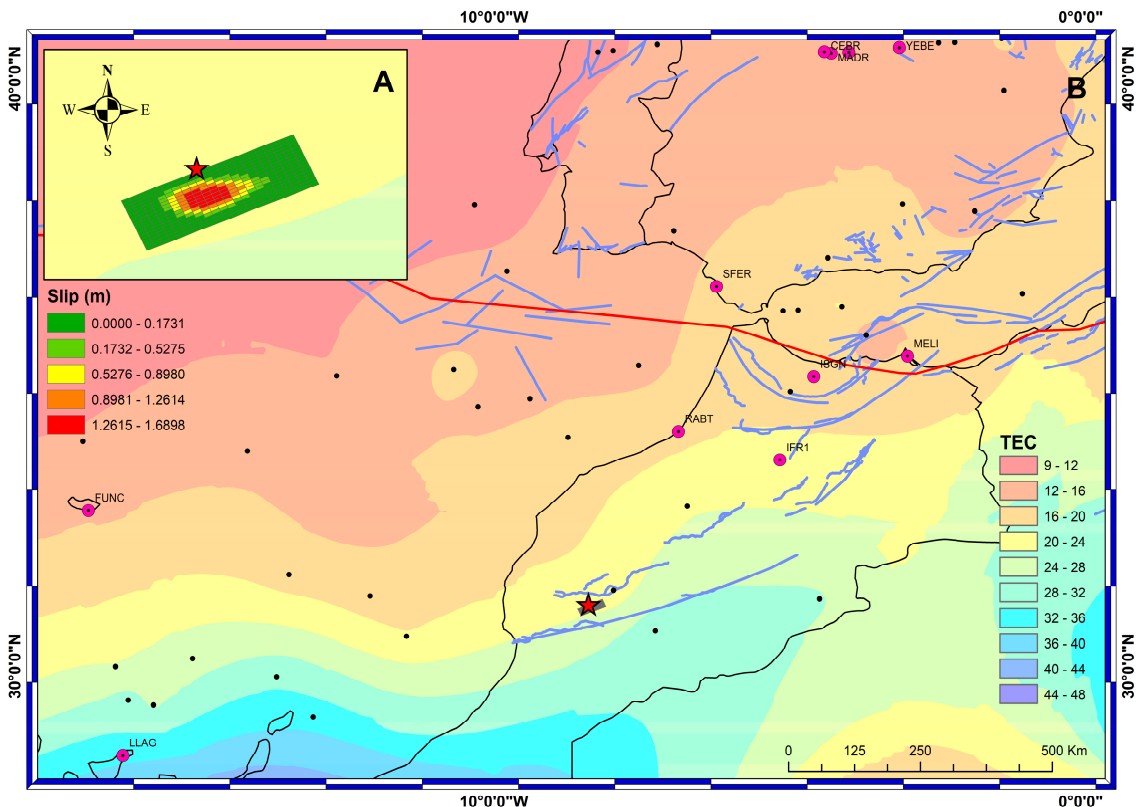

**Figure 8.** Spatial distribution of vTEC as per satellite PRNs, which are illustrated by black circles for the peak negative anomaly time at 18.75 UTC. (**A**) represents the TEC behaviour within the earthquake's epicenter region and the related finite faults. (**B**) represents the TEC behaviour within the seismopreparation zone corresponding to the satellite PRNs.

### *4.3. InSAR Observation*

To conduct InSAR processing in our study, we utilized a dataset comprising 31 satellite images captured by Sentinel-1 during the period from September 2022 to September 2023. Employing the small baseline subset (SBAS) methodology, a technique known for its effectiveness in generating interferograms [3,45], revealed ground displacements triggered by the Moroccan earthquake displaying temporal ground changes. Figure 9 provides an illustration of the interferogram network created during this critical processing step.

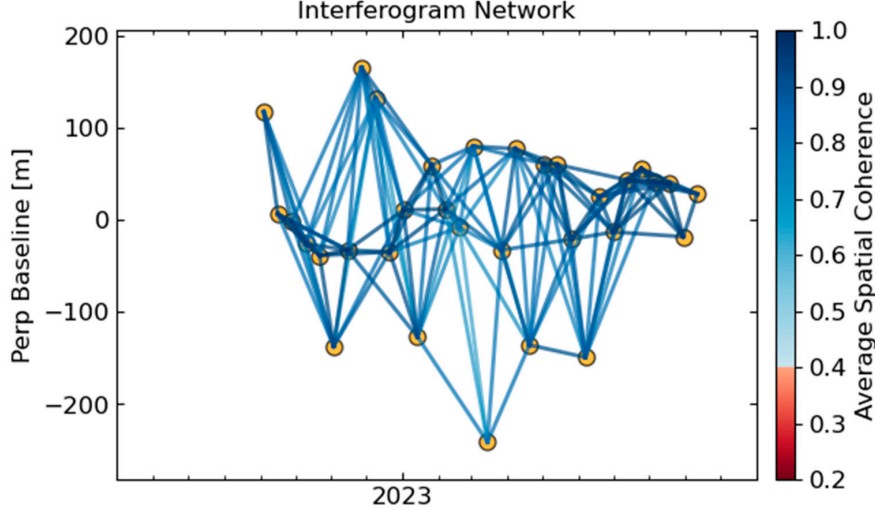

**Figure 9.** Network of SAR interferograms defined using spatial and temporal baseline constraints.

In analyzing the cumulative displacements occurring between September 2022 and August 2023, we observed a notable negative trend, particularly along the line of sight (LOS) direction of the satellite. However, following the earthquake event, this same geographical area exhibited a significant positive cumulative displacement of approximately 15 cm. Figure 10 visually represents these cumulative displacements before and after the earthquake.

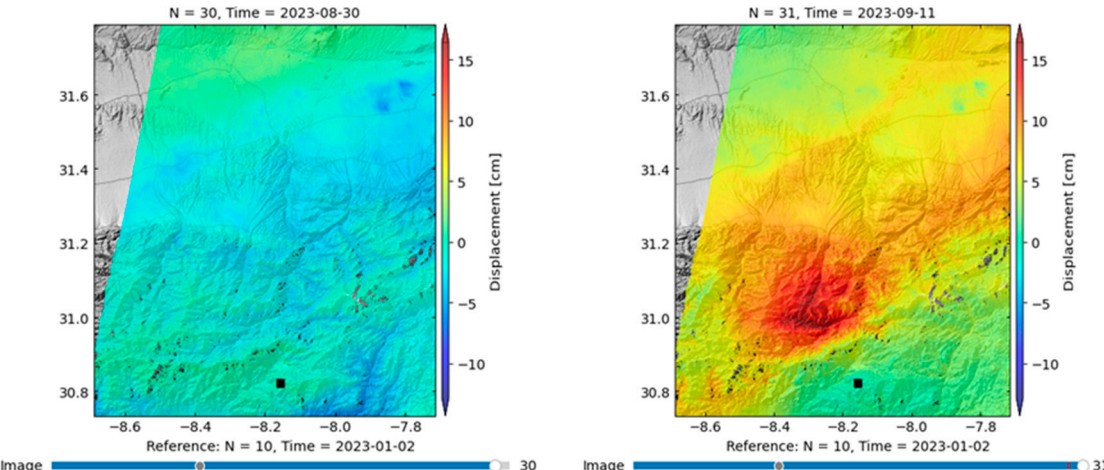

**Figure 10.** Cumulative displacements before and after the earthquake.

Notably, Figure 11 presents a time series focusing on the location with the most pronounced displacements. This time series data reveals a remarkable shift of 15 cm that occurred between 30 August and 11 September, coinciding with the patterns illustrated in Figure 10. Moreover, based on the insights gleaned from this time series, we calculated the velocity of this point to be 4.22 cm per year.

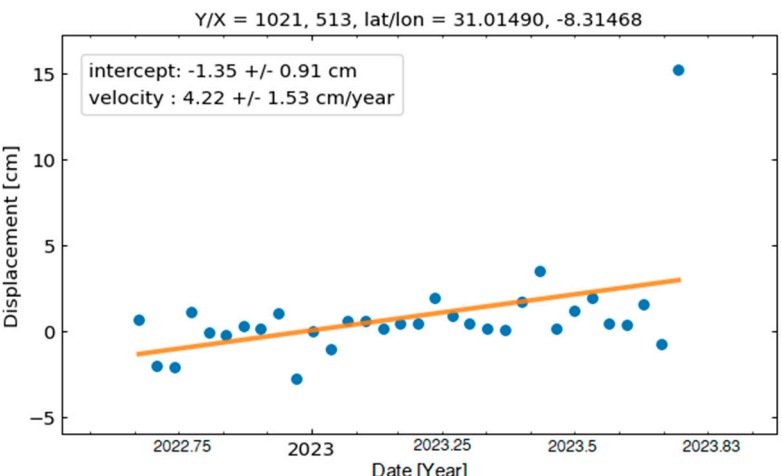

**Figure 11.** Time series of the point with the largest change.

The relationship between these InSAR results and the TEC anomalies observed in our study is noteworthy. The abrupt ground uplift captured by InSAR coincides with the timeframe during which significant positive TEC anomalies were detected leading up to the earthquake. This synchronization suggests a strong correlation between ground displacement and ionospheric disturbances, emphasizing the interconnectedness of tectonical and ionospheric processes in the context of earthquake precursors. Together, these findings underscore the value of combining InSAR and TEC analysis for a more comprehensive understanding of seismic events and their potential early warning indicators. Our multi-faceted approach, incorporating space weather analysis, TEC patterns, and InSAR data,

enriches our comprehension of these geophysical connections, offering essential insights into the LAIC (lithosphere atmosphere ionosphere coupling) mechanism and its relevance in earthquake precursors [46]. These findings contribute to the ongoing exploration of LAIC's potential as a tool for early warning systems and earthquake precursor detection.

## 5. Conclusions

In conclusion, our analysis of space weather conditions and TEC variations leading up to the 6.8 Mw earthquake that occurred on 8 September 2023 has yielded significant insights into the relationship between ionospheric anomalies and seismic events. Our study acknowledges the significance of space weather factors in understanding ionospheric disturbances, emphasizing the potential for solar activity and geomagnetic storms to induce ionospheric irregularities that might be incorrectly attributed to seismic events. It is crucial to acknowledge that while we meticulously evaluated space weather conditions using specific indices (Dst, Kp, and F10.7) to focus exclusively on periods of relative calm, we recognize that Kp < 5 alone is not a sufficient condition to exclude the geomagnetic influence on TEC. This reduced the risk of misinterpreting natural ionospheric variations as earthquake-related anomalies. Our analysis of TEC variations preceding the 6.8 Mw earthquake on 8 September 2023 revealed notable positive and negative anomalies, some occurring up to a week prior to the seismic event. These anomalies, exceeding established thresholds, provided evidence of significant deviations from typical ionospheric conditions. Spatial mapping approaches, considering both station-specific vTEC data and PRNs from multiple GNSS stations, highlighted the correlation between ionospheric anomalies and the earthquake's epicenter. The inclusion of PRNs proved crucial, offering broader coverage and enhanced sensitivity to subtle anomalies. The results of InSAR analysis, in conjunction with the observed TEC anomalies in our study, provide complementary insights into the complex dynamics preceding the Moroccan earthquake. InSAR, as evidenced by the abrupt uplift of the Earth's surface by approximately 15 cm following the seismic event, offers a tangible and spatially explicit representation of ground movements. This uplift aligns with the time frame during which significant positive TEC anomalies were detected, notably up to a week before the earthquake. The convergence of InSAR data and TEC anomalies suggests a strong correlation between the two phenomena. While our research does not directly detect seismic stress, it underscores the temporal relationship between ground uplift, as observed through InSAR, and ionospheric disturbances (TEC anomalies), suggesting an interconnectedness of tectonic and ionospheric processes in the context of earthquake precursors. This synergy between InSAR and TEC data underscores the potential of interdisciplinary approaches in earthquake research, enabling a more comprehensive understanding of the precursory signals associated with seismic events. These findings hold immense promise for enhancing early warning systems and advancing our understanding of potential earthquake precursors. Our study reinforces the notion that ionospheric science is a valuable asset in the ongoing quest to mitigate the impacts of seismic events and protect lives and infrastructure. Looking ahead, future research should explore the application of these methodologies to analyze other large earthquakes and investigate potential correlations between TEC and seismic moment tensors.

**Author Contributions:** Conceptualization, K.N. and C.L.-U.; methodology, K.N.; software, K.N.; validation, R.R.-A., G.S. and M.E.T.-S.; formal analysis, K.N.; investigation, K.N.; resources, G.M.G.-A.; data curation, C.L.-U.; writing—original draft preparation, K.N.; writing—review and editing, K.N.; visualization, G.S.; supervision, R.R.-A.; project administration, M.E.T.-S.; funding acquisition, K.N. All authors have read and agreed to the published version of the manuscript.

**Funding:** This work was carried out with the support (CVU: 1182470) of the National Council of Humanities, Science and Technology (CONAHCyT) in Mexico.

**Data Availability Statement:** The data can be made available upon request.

**Acknowledgments:** The authors extend their appreciation to IGS and NOAA for generously providing access to data. Additionally, the authors are equally thankful to USGS for supplying comprehensive earthquake information. The first author wishes to express his gratitude to Mariana Martínez Mokay for the valuable support extended. Lastly, the authors would like to convey their heartfelt thanks to Leobardo Magallanes-Torres and Jesus Rodolfo Valle for their efforts in providing a conducive and tidy research environment.

**Conflicts of Interest:** The authors declare no conflict of interest.

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
