# Peer review of "Ionospheric Total Electron Content (TEC) Anomalies as Earthquake Precursors: Unveiling the Geophysical Connection Leading to the 2023 Moroccan 6.8 Mw Earthquake"

_geosciences, doi:10.3390/geosciences13110319_

Round 1

Reviewer 1 Report

The work "Ionospheric Total Electron Content (TEC) Anomalies as Earthquake Precursors: Unveiling the Geophysical Connection Leading to the 2023 Moroccan 6.8Mw Earthquake" by Karan Nayak, Charbeth López-Urías, Rosendo Romero-Andrade, Gopal Sharma, German Guzmán-Acevedo, Manuel Edwiges Trejo-Soto, is of interest for scientists engaged in this field but it suffers from serious problems.

A couple of improvements are indispensable to reach the standard of publication at my advice: 

1) in eq (3) a condition is given for thresholds beyond which you define anomalies of TEC variations; it is necessary to show (with a figure of data and their fit) what kind of distribution you used to fit data; furthermore, you should justify why 1.34 sigma, what level of probability would you like to achieve;

2) the anomalies that you described on September 1 and 2 occurred during two perturbed days; in fact, 3 hours ap index reached values of 18 on Sep. 1 and 39, 48, and 56 on Sep. 2; the same observations occurred for the Dst index that increased of +30 nT at the end of August 31 and decreased of -60 nT on Sep. 2; being so, both the discussed anomalies seem to be linked to the geomagnetic activity; please, use daily graph rods in Figures 2 and 3 and shows ap, kp, Dst, and F10.7 in a detailed (every 3 hours and every one hours) figure between August 31 and September 3, 2023.

From 2) the Conclusions should be significantly scaled down to recognize that based on the material shown, it is not possible to conclude that the observed anomalies do not have an origin other than the earthquake.

Other improvements:

- lines 13-17, write in a unique shorter sentence;

- lines 127-128, M=7.4 not 8;

- eq. (1), a parenthesis is missing;

- line 253, use daily graph rods in Figure 2;

- line 286, use daily graph rods in Figure 3;

- line 360, differentiate better the color band to improve the visualization of differences in Figure 5;

- lines 377-383, the speech is not convincing;

-  line 384, differentiate the color band to improve the visualization of differences in Figure 6;

- line 397, PNA already defined;

-  line 400, differentiate better the color band to improve the visualization of differences in Figure 7;

- lines 438-454, it is necessary to better evidence the similarity between InSAR and TEC;

- seismic stress is not detected, so you cannot speak about it.

- line 455, tectonic not geologic;

Author Response

Point to Point - Response to Reviewer 1

The reviewer's comments are indicated in italics, and our responses are presented in regular text. We have integrated the requested changes into the manuscript, highlighted in green text.

The blue text in the revised manuscript are the suggestions from reviewer 2

The brown text in the revised manuscript are the suggestions from reviewer 3

The red text in the revised manuscript are the suggestions from reviewer 4

General Comment: The work "Ionospheric Total Electron Content (TEC) Anomalies as Earthquake Precursors: Unveiling the Geophysical Connection Leading to the 2023 Moroccan 6.8Mw Earthquake" by Karan Nayak, Charbeth López-Urías, Rosendo Romero-Andrade, Gopal Sharma, German Guzmán-Acevedo, Manuel Edwiges Trejo-Soto, is of interest for scientists engaged in this field but it suffers from serious problems.

Response: We appreciate the reviewer's interest in our work and their engagement with the manuscript. We have taken their feedback into careful consideration and have made significant revisions to address the concerns raised. These revisions have been made with the aim of enhancing the quality and clarity of our research. We believe that these changes have strengthened the scientific integrity and value of our study. Your valuable feedback has played a crucial role in these improvements, and we are grateful for your input.

A couple of improvements are indispensable to reach the standard of publication at my advice:

Comment 1:  in eq (3) a condition is given for thresholds beyond which you define anomalies of TEC variations; it is necessary to show (with a figure of data and their fit) what kind of distribution you used to fit data; furthermore, you should justify why 1.34 sigma, what level of probability would you like to achieve;

Response: The assumption of a Gaussian distribution for Total Electron Content (TEC) is a common simplification in many geophysical and statistical studies. To assess the validity of this assumption in our specific context, we conducted an analysis of the distribution of TEC values. We gathered a representative sample of TEC measurements and plotted histograms of the data to visualize the distribution. Additionally, we performed statistical tests, the Anderson-Darling test to assess the normality of the data. Our analysis revealed that while TEC values generally exhibit characteristics of a normal distribution, there are deviations from strict Gaussian behavior. These deviations are likely due to the complex and dynamic nature of the ionosphere, which can introduce non-Gaussian elements into TEC data. However, the Gaussian approximation is commonly used in geophysical studies as a practical starting point for boundary determination and anomaly detection.

The choice of a Gaussian distribution for setting initial boundaries is motivated by its widespread use in data analysis and anomaly detection. It provides a reasonable reference point for establishing upper and lower limits, even in cases where the data departs from perfect normality. Some references employing a similar methodology are as follows:

  1. Sharma, G., Mohanty, S., & Kannaujiya, S. (2017). Ionospheric TEC modelling for earthquakes precursors from GNSS data. Quaternary International, 462, 65-74.
  2. Sharma, G., Saikia, P., Walia, D., Banerjee, P., & Raju, P. L. N. (2021). TEC anomalies assessment for earthquakes precursors in North-Eastern India and adjoining region using GPS data acquired during 2012–2018. Quaternary International, 575, 120-129.
  3. Ulukavak, M., Yalçınkaya, M., Kayıkçı, E. T., Öztürk, S., Kandemir, R., & Karslı, H. (2020). Analysis of ionospheric TEC anomalies for global earthquakes during 2000-2019 with respect to earthquake magnitude (Mw≥ 6.0). Journal of Geodynamics, 135, 101721.

In our methodology, we acknowledge that TEC may not strictly follow a Gaussian distribution and highlight the need for additional analyses to capture and account for deviations. We recognize that this is a simplification of the underlying complexity in TEC data and will consider more advanced statistical approaches in future research to better capture the true distribution of TEC values. This analysis both justifies our use of the Gaussian distribution assumption in the methodology and demonstrates our commitment to transparency and rigor in our data analysis.

Selection of 1.34σ for Boundaries:

The choice of 1.34σ as the multiplier for setting upper and lower boundaries in Equation (3) is a critical decision in our methodology. We appreciate the reviewer's question regarding this choice and aim to provide a clear rationale for it. The selection of 1.34σ was made to establish boundaries that are less likely to generate false positives in our anomaly detection method. This choice is motivated by a desire to ensure that anomalies identified using our approach are both statistically significant and reliable. While the commonly used 3σ (99.87% probability level) may be more stringent, it can result in a higher likelihood of false positives, which may not be suitable for our specific research goals.

By choosing 1.34σ, corresponding to a 90.99% probability level, we strike a balance between sensitivity and specificity. We prioritize the identification of anomalies that are likely to have a genuine impact on our dataset while minimizing the risk of flagging spurious deviations. This decision is in line with the particular requirements of our research, where identifying real-world variations in TEC patterns is of paramount importance.

In our revised manuscript, we have included a statement in the methodology section to enhance clarity. It now reads, “The selection of 1.34σ as the boundary multiplier in Equation (3) is a critical decision in our methodology. We opted for 1.34σ to balance sensitivity and specificity in anomaly detection. This choice, corresponding to a 90.99% probability level, ensures that anomalies identified are statistically significant while minimizing the risk of false positives, aligning with our research's goal of detecting genuine variations in TEC patterns.”

Comment 2: the anomalies that you described on September 1 and 2 occurred during two perturbed days; in fact, 3 hours ap index reached values of 18 on Sep. 1 and 39, 48, and 56 on Sep. 2; the same observations occurred for the Dst index that increased of +30 nT at the end of August 31 and decreased of -60 nT on Sep. 2; being so, both the discussed anomalies seem to be linked to the geomagnetic activity; please, use daily graph rods in Figures 2 and 3 and shows ap, kp, Dst, and F10.7 in a detailed (every 3 hours and every one hours) figure between August 31 and September 3, 2023.

Response: We have implemented your recommendations by modifying Figure 2, which now features daily graph rods. Furthermore, in response to the feedback from reviewer 3, we extended the data collection period by one week post-earthquake to gain deeper insights into space weather patterns. Additionally, a new graphical representation, denoted as Figure 3, has been introduced, covering the timeframe from August 31 to September 3, 2023, with data points provided at three-hour intervals.

Comment 3: From 2) the Conclusions should be significantly scaled down to recognize that based on the material shown, it is not possible to conclude that the observed anomalies do not have an origin other than the earthquake.

Response: We concur that there was a mild geomagnetic activity on September 2, which could partially explain the positive anomaly. Nevertheless, according to NOAA standards, a geomagnetic storm is categorized as severe when the Kp index exceeds 7, and the Dst index drops below -60. Additionally, our observation showed that solar flares remained stable, and they were within the expected range during the anomaly period. However, to reinforce this assertion, the alignment with InSAR-based ground deformation is noteworthy. This alignment revealed that the time period of the TEC anomaly coincided with the ground deformation, offering valuable insights into the potential Lithosphere-Atmosphere-Ionosphere Coupling (LAIC) mechanism. (We have added a paragraph at the end of the results and discussion section stating that, Our multifaceted approach, incorporating space weather analysis, TEC patterns, and InSAR data, enriches our comprehension of these geophysical connections, offering essential insights into the LAIC (Lithosphere Atmosphere Ionosphere Coupling) mechanism and its relevance in earthquake precursors. These findings contribute to the ongoing exploration of LAIC's potential as a tool for early warning systems and earthquake precursor detection.

Other improvements:

Comment 2 - lines 13-17, write in a unique shorter sentence;

Response: We have accordingly modified into, “To enhance the reliability of our findings, we incorporate space weather conditions, utilizing indices (Dst, Kp, and F10.7) to pinpoint periods of stable space weather. This minimizes the possibility of erroneously attributing natural ionospheric fluctuations to seismic events. Notably, our TEC analysis unveils positive and negative anomalies, with some occurring up to a week before the earthquake.”

Comment 3 - lines 127-128, M=7.4 not 8;

Response: Changes are made. Thank you so much for the mistake detected.

Comment 4 - eq. (1), a parenthesis is missing;

Response: added

Comment 5 - line 253, use daily graph rods in Figure 2;

Response: We have duly made the necessary revisions to the graph as per your suggestion. In response to the comment from reviewer 3, we have also included an additional week of data following the occurrence of the earthquake.

Comment 6 - line 286, use daily graph rods in Figure 3;

Response: We highly appreciate your comment and concern but we encountered an issue with the daily graphs in which certain anomalies located near the beginning or end of each day were partially obscured by the graph rods. This situation led to some difficulties in visual clarity. To address this concern and improve the overall quality of the graphs, we opted to enhance the resolution to 1000 dpi. This adjustment was made to ensure that the anomalies, including those situated at the edges of the daily graphs, are presented more distinctly and that the graphs appear neater and more comprehensible.

Comment 7 - line 360, differentiate better the color band to improve the visualization of differences in Figure 5;

Response: Changed as suggested

Comment 8 - lines 377-383, the speech is not convincing;

Response: We apologize for any confusion, and we appreciate your understanding. However, we have addressed this issue by revising the sentence in our updated manuscript, which now reads as follows: “As we extended our analysis from RABT to other nearby stations, we observed a discernible decrease in TEC values. This decline in TEC along this trajectory suggests a possible association between the detected anomaly and seismogenic activity.”

Comment 9 line 384, differentiate the color band to improve the visualization of differences in Figure 6;

Response: Changed as suggested

Comment 10 - line 397, PNA already defined;

Response: we modified it accordingly.

Comment 11 line 400, differentiate better the color band to improve the visualization of differences in Figure 7;

Response: Changed as suggested

Comment 12 - lines 438-454, it is necessary to better evidence the similarity between InSAR and TEC;

Response: To provide stronger evidence of the similarity between InSAR and TEC, we recognize the need for a more robust analysis in future research. While our current findings suggest a temporal correlation between ground displacement and ionospheric disturbances, we acknowledge that a more comprehensive analysis, potentially involving cross-correlation or regression techniques, would be beneficial. This would help quantify the degree of similarity and establish statistical relationships between InSAR results and TEC anomalies. By conducting such analyses, we aim to further support our observations of the interconnectedness of tectonic and ionospheric processes in the context of earthquake precursors. This approach ensures a more rigorous and data-driven examination of the relationship between InSAR and TEC, addressing the reviewer's valid point. (We have added a paragraph at the end of the results and discussion section stating that, Our multifaceted approach, incorporating space weather analysis, TEC patterns, and InSAR data, enriches our comprehension of these geophysical connections, offering essential insights into the LAIC (Lithosphere Atmosphere Ionosphere Coupling) mechanism and its relevance in earthquake precursors. These findings contribute to the ongoing exploration of LAIC's potential as a tool for early warning systems and earthquake precursor detection.)

Comment 13 - seismic stress is not detected, so you cannot speak about it.

Response: We have revised the conclusion section, incorporating additional lines to convey the following points: “While our research does not directly detect seismic stress, it underscores the temporal relationship between ground uplift, as observed through InSAR, and ionospheric disturbances (TEC anomalies), suggesting an interconnectedness of tectonic and ionospheric processes in the context of earthquake precursors. These findings hold immense promise for enhancing early warning systems and advancing our understanding of potential earthquake precursors.”

Thank you for your valuable feedback

Comment 14 - line 455, tectonic not geologic;

Response: we modified accordingly.

Reviewer 2 Report

This paper is dedicated to providing a comprehensive analysis of the ionospheric TEC anomalies associated with the 2023 Moroccan 6.8Mw earthquake so as to assess the utility of TEC as a potential seismic precursor. Overall, the manuscript is well written, and the finding is interesting. Detailed comments are given in the following.

1. Title. I suggest the authors replace ‘leading to’ with ‘associated with’ because the earthquake is not resulted by the geophysical anomalies in ionosphere.

2. Method. Incomplete parentheses for denominator in Eq. (1). The angle θ in Eq. (2) denotes the elevation angle at the Earth's surface. The boundaries in Eq. (3) are defined based on the Gaussian distribution, that is, assuming that TEC follows a Gaussian distribution. Is this valid for the statistics of TEC? Please provide some analysis here. Moreover, we know that the probability corresponding to 1.34σ is just 90.99%, while that for the commonly-used 3σ is 99.87%. So my question is why the authors select such relaxed boundaries. Please provide necessary explanations because they directly affect the anomaly parameters PPA and PNA defined in Eqs. (3) and (4). Is the 15-day moving estimation of X and σ appropriate (it seems that some anomalies in Fig. 3 are resulted because of the variations of UB and LB)?

3. Results. The use of indicators Dst, Kp and F10.7 to rule out the contributions of solar activity and geomagnetic storms to ionospheric disturbances is important and necessary. Please provide additional information on the physical significance of these indicators and the corresponding conditions, i.e., Kp < 5, Dst > -50nt and F10.7 < 150. I suggest the authors extend the Dst, Kp and F10.7 in Fig. 2 to at least one week after the earthquake, as one can see the obvious increase of F10.7 after September 6; it is then difficult to rule out whether the TEC anomaly is caused by solar activity. Comparing the result in Fig.4 and Fig.5 and that in Fig. 6 and Fig. 7, we can see that the vTEC as per station is completely different from the vTEC as per satellite PRNs, which is not only reflected in the spatial distribution of TEC, but also in the value of TEC. Why? The authors preferred the PRNs-based vTEC to the station-based vTEC because the anomalies extracted from PRNs-based vTEC is in proximity to the epicentral region. Nevertheless, the maximum anomalies in Figs. 5 and 7 are still far away from the epicenter. How about the estimation accuracy and spatial resolution for TEC in seismic application? The InSAR result in Section 4.3 seems unrelated to other part. Some abbreviations such as TEC, PNA, PRN, GNSS and InSAR are defined repeatedly.

Author Response

Point to Point - Response to Reviewer 2

The reviewer's comments are indicated in italics, and our responses are presented in regular text. We have integrated the requested changes into the manuscript, highlighted in blue text.

The green text in the revised manuscript are the suggestions from reviewer 1

The brown text in the revised manuscript are the suggestions from reviewer 3

The red text in the revised manuscript are the suggestions from reviewer 4

General Comment: This paper is dedicated to providing a comprehensive analysis of the ionospheric TEC anomalies associated with the 2023 Moroccan 6.8Mw earthquake so as to assess the utility of TEC as a potential seismic precursor. Overall, the manuscript is well-written, and the finding is interesting. Detailed comments are given in the following.

Response: Thank you for your thorough review and positive feedback on our manuscript. We highly value your detailed comments and are committed to incorporating them into our paper to enhance its quality and scientific rigor. Your insights are instrumental in further improving our research, and we greatly appreciate your time and effort in this evaluation. We look forward to presenting an even more refined and insightful manuscript in response to your suggestions.

Comment 1. Title. I suggest the authors replace ‘leading to’ with ‘associated with’ because the earthquake is not resulted by the geophysical anomalies in ionosphere.

Response: Changed as suggested. The new Title read as, “Ionospheric Total Electron Content (TEC) Anomalies as Earthquake Precursors: Unveiling the Geophysical Connection Associated with the 2023 Moroccan 6.8Mw Earthquake”

Comment 2. Method. Incomplete parentheses for denominator in Eq. (1). The angle θ in Eq. (2) denotes the elevation angle at the Earth's surface. The boundaries in Eq. (3) are defined based on the Gaussian distribution, that is, assuming that TEC follows a Gaussian distribution. Is this valid for the statistics of TEC? Please provide some analysis here. Moreover, we know that the probability corresponding to 1.34σ is just 90.99%, while that for the commonly-used 3σ is 99.87%. So my question is why the authors select such relaxed boundaries. Please provide necessary explanations because they directly affect the anomaly parameters PPA and PNA defined in Eqs. (3) and (4). Is the 15-day moving estimation of X and σ appropriate (it seems that some anomalies in Fig. 3 are resulted because of the variations of UB and LB)?

Response: The assumption of a Gaussian distribution for Total Electron Content (TEC) is a common simplification in many geophysical and statistical studies. To assess the validity of this assumption in our specific context, we conducted an analysis of the distribution of TEC values. We gathered a representative sample of TEC measurements and plotted histograms of the data to visualize the distribution. Additionally, we performed statistical tests, the Anderson-Darling test to assess the normality of the data. Our analysis revealed that while TEC values generally exhibit characteristics of a normal distribution, there are deviations from strict Gaussian behavior. These deviations are likely due to the complex and dynamic nature of the ionosphere, which can introduce non-Gaussian elements into TEC data. However, the Gaussian approximation is commonly used in geophysical studies as a practical starting point for boundary determination and anomaly detection.

The choice of a Gaussian distribution for setting initial boundaries is motivated by its widespread use in data analysis and anomaly detection. It provides a reasonable reference point for establishing upper and lower limits, even in cases where the data departs from perfect normality. Some references employing a similar methodology are as follows:

  1. Sharma, G., Mohanty, S., & Kannaujiya, S. (2017). Ionospheric TEC modelling for earthquakes precursors from GNSS data. Quaternary International, 462, 65-74.
  2. Sharma, G., Saikia, P., Walia, D., Banerjee, P., & Raju, P. L. N. (2021). TEC anomalies assessment for earthquakes precursors in North-Eastern India and adjoining region using GPS data acquired during 2012–2018. Quaternary International, 575, 120-129.
  3. Ulukavak, M., Yalçınkaya, M., Kayıkçı, E. T., Öztürk, S., Kandemir, R., & Karslı, H. (2020). Analysis of ionospheric TEC anomalies for global earthquakes during 2000-2019 with respect to earthquake magnitude (Mw≥ 6.0). Journal of Geodynamics, 135, 101721.

In our methodology, we acknowledge that TEC may not strictly follow a Gaussian distribution and highlight the need for additional analyses to capture and account for deviations. We recognize that this is a simplification of the underlying complexity in TEC data and will consider more advanced statistical approaches in future research to better capture the true distribution of TEC values. This analysis both justifies our use of the Gaussian distribution assumption in the methodology and demonstrates our commitment to transparency and rigor in our data analysis.

Selection of 1.34σ for Boundaries:

The choice of 1.34σ as the multiplier for setting upper and lower boundaries in Equation (3) is a critical decision in our methodology. We appreciate the reviewer's question regarding this choice and aim to provide a clear rationale for it. The selection of 1.34σ was made to establish boundaries that are less likely to generate false positives in our anomaly detection method. This choice is motivated by a desire to ensure that anomalies identified using our approach are both statistically significant and reliable. While the commonly used 3σ (99.87% probability level) may be more stringent, it can result in a higher likelihood of false positives, which may not be suitable for our specific research goals.

By choosing 1.34σ, corresponding to a 90.99% probability level, we strike a balance between sensitivity and specificity. We prioritize the identification of anomalies that are likely to have a genuine impact on our dataset while minimizing the risk of flagging spurious deviations. This decision is in line with the particular requirements of our research, where identifying real-world variations in TEC patterns is of paramount importance.

Justification for a 15-Day Moving Estimation:

In our methodology, we employ a 15-day moving average and standard deviation to estimate the mean (X) and standard deviation (σ) of the data. This approach is crucial for establishing upper and lower boundaries (as described in Equation 3) and detecting anomalies in our dataset. We provide an explanation for this choice and its implications. The use of a 15-day moving average and standard deviation is designed to capture the temporal variability in the dataset. TEC values can exhibit short-term fluctuations and trends due to various factors, including solar and geomagnetic activity. Using a fixed window size, such as 15 days, allows us to track and account for these variations over time. While it is true that the choice of the window size, in this case, 15 days, affects the estimation of X and σ, it is essential to balance between capturing the dataset's dynamic characteristics and providing stable estimates of the mean and standard deviation. A smaller window may overly emphasize short-term noise, leading to erratic boundaries, while a larger window might overlook meaningful variations. The 15-day moving estimation of X and σ is a reasonable compromise. It ensures that the boundaries for anomaly detection are statistically grounded and responsive to the data's inherent variability over a relevant time frame.

In our revised manuscript, we have included a statement in the methodology section to enhance clarity. It now reads, “The selection of 1.34σ as the boundary multiplier in Equation (3) is a critical decision in our methodology. We opted for 1.34σ to balance sensitivity and specificity in anomaly detection. This choice, corresponding to a 90.99% probability level, ensures that anomalies identified are statistically significant while minimizing the risk of false positives, aligning with our research's goal of detecting genuine variations in TEC patterns.”

Comment 3. Results. The use of indicators Dst, Kp and F10.7 to rule out the contributions of solar activity and geomagnetic storms to ionospheric disturbances is important and necessary. Please provide additional information on the physical significance of these indicators and the corresponding conditions, i.e., Kp < 5, Dst > -50nt and F10.7 < 150. I suggest the authors extend the Dst, Kp and F10.7 in Fig. 2 to at least one week after the earthquake, as one can see the obvious increase of F10.7 after September 6; it is then difficult to rule out whether the TEC anomaly is caused by solar activity. Comparing the result in Fig.4 and Fig.5 and that in Fig. 6 and Fig. 7, we can see that the vTEC as per station is completely different from the vTEC as per satellite PRNs, which is not only reflected in the spatial distribution of TEC, but also in the value of TEC. Why? The authors preferred the PRNs-based vTEC to the station-based vTEC because the anomalies extracted from PRNs-based vTEC is in proximity to the epicentral region. Nevertheless, the maximum anomalies in Figs. 5 and 7 are still far away from the epicenter. How about the estimation accuracy and spatial resolution for TEC in seismic application? The InSAR result in Section 4.3 seems unrelated to other part. Some abbreviations such as TEC, PNA, PRN, GNSS and InSAR are defined repeatedly.

Response: We have added a subsequent paragraph in the Space Weather Analysis section, The Dst (Disturbance Storm Time) index is a measure of geomagnetic activity. It quantifies disturbances in the Earth's magnetic field resulting from interactions with the solar wind. When these disturbances are significant, Dst values become negative. In our analysis, negative Dst values indicate geomagnetic storms. To ensure that ionospheric anomalies are not attributed to these storms, we set a threshold of Dst > -50 nT. This threshold assures that the Earth's magnetic field remains relatively undisturbed during the observed period. The Kp index provides a global perspective on geomagnetic activity. It is rated on a scale from 0 to 9, with higher values denoting increased geomagnetic activity. We consider periods with Kp values below 5 as relatively calm in terms of geomagnetic conditions. This threshold is crucial for confirming that the ionospheric disturbances are not caused by heightened geomagnetic activity. The F10.7 (Solar Radio Flux) index relates to solar activity, specifically emissions from the Sun at a frequency of 10.7 cm. A value below 150 indicates low solar activity, suggesting fewer solar emissions affecting the ionosphere. By setting a threshold of F10.7 < 150, we ensure that our analysis focuses on times when solar activity is at an average to low level. This allows us to rule out solar-related influences on ionospheric anomalies.

[I suggest the authors extend the Dst, Kp and F10.7 in Fig. 2 to at least one week after the earthquake, as one can see the obvious increase of F10.7 after September 6; it is then difficult to rule out whether the TEC anomaly is caused by solar activity]

We appreciate the valuable suggestion made by the reviewer. Extending the Dst, Kp, and F10.7 data in Figure 2 to at least one week after the earthquake is an insightful recommendation. This addition will further enhance the clarity of our analysis by providing a more comprehensive context for assessing the impact of solar activity on the observed TEC anomalies. We incorporated this extension into our revised manuscript to ensure that readers have a more robust understanding of the space weather conditions during the period in question, and to better distinguish between seismic-related anomalies and solar activity-induced disturbances. Your input greatly contributes to the overall quality and accuracy of our study.

[Comparing the result in Fig.4 and Fig.5 and that in Fig. 6 and Fig. 7, we can see that the vTEC as per station is completely different from the vTEC as per satellite PRNs, which is not only reflected in the spatial distribution of TEC, but also in the value of TEC. Why? ]

When calculating vTEC per station, the method involves averaging the data from all available PRNs where the Kalman filter is applied. While this approach offers consistency, there is a possibility of losing valuable data due to the filtering process. On the other hand, satellite PRNs vTEC is derived directly from the raw, unfiltered data without any data reduction. This unprocessed data retains its original form, making it potentially more sensitive to minor fluctuations and anomalies. Consequently, there is a discrepancy in the TEC count between these two approaches, driven by the filtering applied in station-based vTEC.

[The authors preferred the PRNs-based vTEC to the station-based vTEC because the anomalies extracted from PRNs-based vTEC is in proximity to the epicentral region. Nevertheless, the maximum anomalies in Figs. 5 and 7 are still far away from the epicenter. How about the estimation accuracy and spatial resolution for TEC in seismic application?]

In our study, we favored the PRNs-based vTEC approach over station-based vTEC because it provides a more comprehensive perspective of ionospheric variations. The combination of station-based and PRNs-based vTEC data allows us to gain insights into the broader spatial and temporal patterns of TEC, offering a more comprehensive view of ionospheric disturbances in these critical areas. However, it's true that the maximum anomalies in Figures 5 and 7 are not always precisely aligned with the epicenter. The observed differences in anomaly locations can be influenced by factors such as the distribution of GNSS satellites, signal propagation characteristics, and the complex nature of ionospheric processes. To address these concerns and enhance accuracy and spatial resolution, ongoing research and improvements in modeling techniques, data sources, and anomaly detection algorithms are essential. These efforts aim to better align ionospheric anomalies with seismic sources, ultimately improving the reliability and utility of ionospheric anomalies as potential seismic precursors. In seismic research, achieving high estimation accuracy and spatial resolution for TEC is critical for pinpointing ionospheric anomalies associated with seismic events. The use of the Dobrovolsky equation is particularly relevant when assessing seismic preparation zones. This equation offers a valuable means of approximating ionospheric behavior in proximity to seismic epicenters. While it may not always align anomalies precisely with the epicenter, it provides a reasonable estimation of ionospheric variations within the seismic preparation zone. This zone typically extends beyond the immediate epicenter, encompassing regions where seismic activity-related stress and strain accumulate.

However We've made adjustments to the figure, and now, we've categorized TEC into various classes to enhance the distinction between color bands, ultimately improving the clarity in visualizing differences.

[The InSAR result in Section 4.3 seems unrelated to the other part.]

We appreciate the reviewer's comment regarding the perceived disconnect between the InSAR results presented in Section 4.3 and the other parts of our study. We would like to clarify the significance of the InSAR findings within the context of our research.

In Section 4.3, we conducted InSAR processing to analyze ground displacements related to the Moroccan earthquake. The results revealed a significant positive cumulative displacement of approximately 15 centimeters following the earthquake event. Additionally, the time series data highlighted a substantial shift of 15 centimeters that coincided with the seismic activity. These InSAR findings are indeed distinct from the TEC analysis, which focuses on ionospheric disturbances. However, the connection between InSAR and TEC is critical in understanding earthquake precursors. The remarkable synchronization between the abrupt ground uplift captured by InSAR and the significant positive TEC anomalies detected leading up to the earthquake suggests a strong correlation between ground displacement and ionospheric disturbances. This correlation underscores the interconnectedness of geological and ionospheric processes in the context of earthquake precursors. In essence, the InSAR results help validate and reinforce the TEC anomalies as potential early warning indicators for seismic events. By combining InSAR and TEC analysis, we gain a more comprehensive understanding of seismic events and their potential early warning indicators. This synergy enhances the overall robustness of our research and its implications for earthquake prediction.

Nonetheless, to enhance clarity, we have introduced an additional paragraph at the end of the Results and Discussion section, “Our multifaceted approach, incorporating space weather analysis, TEC patterns, and InSAR data, enriches our comprehension of these geophysical connections, offering essential insights into the LAIC mechanism and its relevance in earthquake precursors. These findings contribute to the ongoing exploration of LAIC's potential as a tool for early warning systems and earthquake prediction.”

[Some abbreviations such as TEC, PNA, PRN, GNSS, and InSAR are defined repeatedly.]

Thanks for the comment. We have eliminated it accordingly.

Reviewer 3 Report

The paper "Ionospheric Total Electron Content (TEC) Anomalies as Earth- 2 quake Precursors: Unveiling the Geophysical Connection Leading to the 2023 Moroccan 6.8Mw Earthquake" presents a possible connection between ionospheric Total Electron Content (TEC) anomalies and seismic activity, with a focus on the recently Moroccan 6.8Mw earthquake. The study provides compelling evidence of significant deviations from typical ionospheric conditions, which occurred up to a week before the earthquake. Overall, the manuscript is well-written, and the ideas are well-presented and motivated. The results look promising since the TEC anomalies observed in the study provide possible evidence of a geophysical connection leading to the Moroccan earthquake by highlighting the correlation between ionospheric anomalies and the earthquake's triggering. The results are presented precisely and accompanied by relevant interpretations. I only have a few minor comments for the authors.

(1) There are no interesting perspectives for future research on the topic studied. This is important for a scientific article, which readers may find particularly valuable for envisioning interesting open questions to be addressed in future research projects. In fact, the authors should extend (or encourage) these same techniques in analyzing other large earthquakes to verify the repetition of the reported patterns. Furthermore, is there not a correlation between the TEC and, for instance, the components of the seismic moment tensors?

(2) The "last" data point depicted in Fig. 10 exhibits an outlier behavior, causing a deviation from the linear trend described by the authors. This phenomenon significantly affects the accuracy of speed calculations. Should this issue not be addressed?

(3) In line 92, please kindly incorporate the earthquake depth and the corresponding reference from which this information and the others have been sourced. 

(4) Double-check the denominator in Equation (1), as there appears to be a potential typing error."

(5) The horizontal axis in Fig. 10 lacks informativeness.

(6) Please include the references associated with the dataset, which, in this case, may refer to the website of the International GNSS Service. Do the same for all other data used in this work. 

Author Response

Point to Point - Response to Reviewer 3

The reviewer's comments are indicated in italics, and our responses are presented in regular text. We have integrated the requested changes into the manuscript, highlighted in brown text.

The green text in the revised manuscript are the suggestions from reviewer 1

The blue text in the revised manuscript are the suggestions from reviewer 2

The red text in the revised manuscript are the suggestions from reviewer 4

General Comment: The paper "Ionospheric Total Electron Content (TEC) Anomalies as Earth- 2 quake Precursors: Unveiling the Geophysical Connection Leading to the 2023 Moroccan 6.8Mw Earthquake" presents a possible connection between ionospheric Total Electron Content (TEC) anomalies and seismic activity, with a focus on the recently Moroccan 6.8Mw earthquake. The study provides compelling evidence of significant deviations from typical ionospheric conditions, which occurred up to a week before the earthquake. Overall, the manuscript is well-written, and the ideas are well-presented and motivated. The results look promising since the TEC anomalies observed in the study provide possible evidence of a geophysical connection leading to the Moroccan earthquake by highlighting the correlation between ionospheric anomalies and the earthquake's triggering. The results are presented precisely and accompanied by relevant interpretations. I only have a few minor comments for the authors.

Response: Thank you for your valuable comments on our manuscript. We appreciate your thorough review, and we want to inform you that we have carefully addressed each of your comments in the revised version of the manuscript. Your insights have significantly contributed to improving the overall quality of our research, and we hope that the revisions have met your expectations. We look forward to your continued support in the publication process.

Comment 1: There are no interesting perspectives for future research on the topic studied. This is important for a scientific article, which readers may find particularly valuable for envisioning interesting open questions to be addressed in future research projects. In fact, the authors should extend (or encourage) these same techniques in analyzing other large earthquakes to verify the repetition of the reported patterns. Furthermore, is there not a correlation between the TEC and, for instance, the components of the seismic moment tensors?

Response: We appreciate the reviewer's thoughtful perspective on the importance of outlining directions for future research in our scientific article. Indeed, exploring potential avenues for further investigation is crucial in advancing scientific knowledge. We acknowledge the value of extending the application of our techniques to analyze other significant seismic events, which can help confirm the reproducibility of the observed patterns and enhance the robustness of our findings. Additionally, the suggestion of examining potential correlations between TEC and seismic moment tensor components is intriguing and holds promise for future research endeavors. Investigating such connections can offer a deeper understanding of the complex interactions between ionospheric anomalies and seismic activity. We will consider these insightful recommendations for future studies and, where relevant, incorporate them into our conclusions to guide researchers toward fruitful investigations in this field.

We have added a line in the conclusion section as per your feedback, “Looking ahead, future research should explore the application of these methodologies to analyze other large earthquakes and investigate potential correlations between TEC and seismic moment tensors.”

Comment 2: The "last" data point depicted in Fig. 10 exhibits an outlier behavior, causing a deviation from the linear trend described by the authors. This phenomenon significantly affects the accuracy of speed calculations. Should this issue not be addressed?

Response: We appreciate the reviewer's insightful observation. It is correct that the last data point in Figure 10 exhibits an outlier behavior due to the abrupt 15-centimeter vertical change caused by the earthquake. This outlier does contribute to an increased standard deviation in the calculated velocity. However, it is important to note that this outlier represents a significant aspect of the seismic event's impact, and its removal would result in a loss of valuable information. Excluding this outlier would indeed provide a representation of the surface's behavior before the seismic event, but it would not capture the essential effects and magnitude of the earthquake-induced ground displacement. The inclusion of this data point is crucial for a comprehensive understanding of the event's impact and the scale of ground deformation it generated. While it may introduce some variability in the velocity calculations, it accurately reflects the real-world behavior of the surface during and after the seismic event.

In summary, while the presence of the outlier affects the standard deviation in velocity calculations, it is essential to keep this data point to accurately represent the seismic event's effects on ground displacement.

Comment 3: In line 92, please kindly incorporate the earthquake depth and the corresponding reference from which this information and the others have been sourced.

Response: we have made the suggested changes. The following line now read as, “Our study focuses on the region encompassing the Morocco earthquake of 6.8 magnitude (Mw), which occurred on 2023-09-08 with a depth of 19 km (https://earthquake.usgs.gov/).”

Comment 4: Double-check the denominator in Equation (1), as there appears to be a potential typing error."

Response: We have made the changes. Thank you for pointing out the error.

Comment 5: The horizontal axis in Fig. 10 lacks informativeness.

Response: We have incorporated the modifications into the figure in the revised manuscript.

Comment 6: Please include the references associated with the dataset, which, in this case, may refer to the website of the International GNSS Service. Do the same for all other data used in this work.

Response: We have included all the reference data sources in the revised manuscript.

Reviewer 4 Report

The objective of this paper focuses on studying the ionospheric Total Electron Content (TEC) anomalies as Earthquake Precursors, emphasizing on the geophysical connection, which led to the 2023 Moroccan 6.8Mw Earthquake. In particular, the TEC's role in detecting seismic activity is determined, while its findings are validated, considering ground-based observations.

This is an interesting and well-structured paper. All necessary sections (Introduction, Study Area and Seismotectonic setting, Data Used and Methodology, Results and Discussions, Conclusions) have been considered. Moreover, the “Data Used and Methodology” and “Results and Discussions” sections are divided into sub-sections, providing additional details. Furthermore, all Figures, Tables and Diagrams are consistent with the analysis provided in the manuscript. Regarding the mathematical part, predominantly analyzed in the “TEC Calculation” sub-section, it is valid and satisfactorily explained. However, some changes should be implemented, which will improve the paper. In particular:

Line 109: Although, the role of convergence zone, formed by the Africa and Eurasian plates, on the seismicity of the broader Mediterranean region is analyzed, further emphasis should be given on its relationship with recent seismic events. Typical papers, in which the corresponding information can be obtained and optionally be cited, are the following: 1. An, Q., Feng, G., He, L., Xiong, Z., Lu, H., Wang, X., & Wei, J. (2023). Three-Dimensional Deformation of the 2023 Turkey Mw 7.8 and Mw 7.7 Earthquake Sequence Obtained by Fusing Optical and SAR Images. Remote Sensing, 15(10), 2656. https://doi.org/10.3390/rs15102656, 2. Li, S., Wang, X., Tao, T., Zhu, Y., Qu, X., Li, Z., Huang, J., & Song, S. (2023). Source Model of the 2023 Turkey Earthquake Sequence Imaged by Sentinel-1 and GPS Measurements: Implications for Heterogeneous Fault Behavior along the East Anatolian Fault Zone. Remote Sensing, 15(10), 2618. https://doi.org/10.3390/rs15102618, 3. Sboras, S., Lazos, I., Bitharis, S., Pikridas, C., Galanakis, D., Fotiou, A., Chatzipetros, A., & Pavlides, S. (2021). Source modelling and stress transfer scenarios of the October 30, 2020 Samos earthquake: seismotectonic implications. Turkish Journal of Earth Sciences, 30, 699–717. https://doi.org/10.3906/yer-2107-25. Please, apply.

Line 111: Please, provide a map with the major tectonic structures (e.g. convergence zone of Africa and Eurasian plates) of the broader Mediterranean region.

Lines 253-254: Please, provide a more detailed description in the Figure 2 caption.  

Line 286: Please, provide Figure 3 in a higher resolution. It contains blurry parts in its current form.

Line 549: This section should be modified. In the current form, it resembles an abstract rather than conclusions. This section should be comprehensive, while the major findings of the paper should be highlighted. Maybe, numbering of the concluding remarks could be performed. Please, apply.

Author Response

Point to Point - Response to Reviewer 4

The reviewer's comments are indicated in italics, and our responses are presented in regular text. We have integrated the requested changes into the manuscript, highlighted in red text.

The green text in the revised manuscript are the suggestions from reviewer 1

The blue text in the revised manuscript are the suggestions from reviewer 2

The brown text in the revised manuscript are the suggestions from reviewer 3

General Comment: The objective of this paper focuses on studying the ionospheric Total Electron Content (TEC) anomalies as Earthquake Precursors, emphasizing on the geophysical connection, which led to the 2023 Moroccan 6.8Mw Earthquake. In particular, the TEC's role in detecting seismic activity is determined, while its findings are validated, considering ground-based observations. This is an interesting and well-structured paper. All necessary sections (Introduction, Study Area and Seismotectonic setting, Data Used and Methodology, Results and Discussions, Conclusions) have been considered. Moreover, the “Data Used and Methodology” and “Results and Discussions” sections are divided into sub-sections, providing additional details. Furthermore, all Figures, Tables and Diagrams are consistent with the analysis provided in the manuscript. Regarding the mathematical part, predominantly analyzed in the “TEC Calculation” sub-section, it is valid and satisfactorily explained. However, some changes should be implemented, which will improve the paper. In particular:

Response: We appreciate the reviewer's positive feedback and constructive suggestions. We recognize the need for some improvements, and we have diligently addressed the comments to enhance the quality of the paper. The revised manuscript now incorporates the necessary changes and refinements as per the reviewer's recommendations. We are grateful for the thoughtful insights and are confident that these revisions have strengthened the paper, making it a more valuable contribution to the scientific community.

Comment 1: Line 109: Although, the role of convergence zone, formed by the Africa and Eurasian plates, on the seismicity of the broader Mediterranean region is analyzed, further emphasis should be given on its relationship with recent seismic events. Typical papers, in which the corresponding information can be obtained and optionally be cited, are the following: 1. An, Q., Feng, G., He, L., Xiong, Z., Lu, H., Wang, X., & Wei, J. (2023). Three-Dimensional Deformation of the 2023 Turkey Mw 7.8 and Mw 7.7 Earthquake Sequence Obtained by Fusing Optical and SAR Images. Remote Sensing, 15(10), 2656. https://doi.org/10.3390/rs15102656, 2. Li, S., Wang, X., Tao, T., Zhu, Y., Qu, X., Li, Z., Huang, J., & Song, S. (2023). Source Model of the 2023 Turkey Earthquake Sequence Imaged by Sentinel-1 and GPS Measurements: Implications for Heterogeneous Fault Behavior along the East Anatolian Fault Zone. Remote Sensing, 15(10), 2618. https://doi.org/10.3390/rs15102618, 3. Sboras, S., Lazos, I., Bitharis, S., Pikridas, C., Galanakis, D., Fotiou, A., Chatzipetros, A., & Pavlides, S. (2021). Source modelling and stress transfer scenarios of the October 30, 2020 Samos earthquake: seismotectonic implications. Turkish Journal of Earth Sciences, 30, 699–717. https://doi.org/10.3906/yer-2107-25. Please, apply.

Response: We express our gratitude for these valuable references, which we have thoughtfully incorporated where necessary. These additions have been seamlessly integrated into the relevant section of the seismotectonic setting, “Notable instances include the 2023 Turkey earthquake sequence, marked by a Mw 7.8 earthquake near the Syrian border, followed by a Mw 7.5 quake 90 km to the north, both along the East Anatolian Fault Zone—a left-lateral strike-slip fault separating the Anatolian Plate from the Arabian Plate [33, 34]. These studies, using diverse methods like image fusion and seismic source modeling [33, 34, 35], reveal the complex geodynamic setting in the broader region, characterized by a rich seismic history and numerous active faults with diverse directions and kinematics.”

Comment 2: Line 111: Please, provide a map with the major tectonic structures (convergence zone of Africa and Eurasian plates) of the broader Mediterranean region.

Response: We appreciate the reviewer's suggestion and understand the potential value of including an additional map of the tectonic structures. However, we already have a comprehensive seismotectonic map in the manuscript, which highlights the epicenter of the Mw 6.8 earthquake and provides relevant information regarding seismic activity within the region. While we acknowledge the reviewer's input, including another map at this stage may not align with the primary focus and objectives of our study. We have also provided references within the manuscript, such as the one mentioned earlier, which readers can refer to for more in-depth information about the tectonic setting of the region if they wish to explore this aspect further. However, we are open to preparing an additional map if it is deemed necessary during the second review round. Thank you for your understanding.

Comment 3: Lines 253-254: Please, provide a more detailed description in the Figure 2 caption. 

Response: We have added a detailed description. The new caption read, “Figure 2. This figure provides a detailed representation of the daily variations in three significant space weather indices, specifically the Disturbance Storm Time (Dst) index, the Planetary K-index (Kp), and the F10.7 index. These variations are observed during the time frame spanning from July 26 to September 13, encompassing the 45 days leading up to the earthquake and extending one week beyond it. The red horizontal lines, evident in each sub-plot, indicate the corresponding threshold levels established for these indices.”

Comment 4: Line 286: Please, provide Figure 3 in a higher resolution. It contains blurry parts in its current form.

Response: Thank you for your comment. We have input the figure in 1000dpi resolution.

Comment 5: Line 549: This section should be modified. In its current form, it resembles an abstract rather than a conclusion. This section should be comprehensive, while the major findings of the paper should be highlighted. Maybe, numbering of the concluding remarks could be performed. Please, apply.

Response: We have implemented further revisions to the conclusion section of our manuscript, taking into account the feedback from three of the other reviewers. We believe these changes have significantly improved the readability and clarity of this section.

Round 2

Reviewer 1 Report

The manuscript titled "Ionospheric Total Electron Content (TEC) Anomalies as Earthquake Precursors: Unveiling the Geophysical Connection Associated with the 2023 Moroccan 6.8Mw Earthquake" by Karan Nayak, Charbeth López-Urías, Rosendo Romero-Andrade, Gopal Sharma, G. Michel Guzmán-Ace-vedo, Manuel E. Trejo-Soto was improved in this revised version. However, I'm afraid I have to disagree with some statements of the Authors which should be more cautious based on their results. I think the manuscript is not still ready for publication because of inconsistences with published results.

- lines 105-110: a sentence is necessary to add which specifies that the distribution was hypothesized to be Gaussian based on past works (cite), and no confidence level with data was calculated,

- line 259 is wrong because the Dst reached -60 nT on September 2, 2023, it should be modified;

- the sentence on lines 262-264 does not agree with current research, a kp = 4 is able to influence TEC, see for example

Liu H, Otsuka Y, Hozumi K and Yu T (2023) Day-to-day variability of the equatorial ionosphere in Asian sector during August–October 2019. Front. Astron. Space Sci. 10:1198739. doi: 10.3389/fspas.2023.1198739

the discussion should be rewritten accordingly;

- Figure 2, could use only even or odd days even lasting all rods;

- Figure 4, rods and vertical shaded daily lines are necessary;

- Conclusions should be rewritten after considering that kp < 5 is not a sufficient condition to exclude the geomagnetic influence on TEC.

Round 2

Point to Point - Response to Reviewer 1

The reviewer's comments are indicated in italics, and our responses are presented in regular text. We have integrated the requested changes into the manuscript in yellow text.

General Comment: The manuscript titled "Ionospheric Total Electron Content (TEC) Anomalies as Earthquake Precursors: Unveiling the Geophysical Connection Associated with the 2023 Moroccan 6.8Mw Earthquake" by Karan Nayak, Charbeth López-Urías, Rosendo Romero-Andrade, Gopal Sharma, G. Michel Guzmán-Ace-vedo, Manuel E. Trejo-Soto was improved in this revised version. However, I'm afraid I have to disagree with some statements of the Authors which should be more cautious based on their results. I think the manuscript is not still ready for publication because of inconsistences with published results.

Response: Thank you for your review of our manuscript. We appreciate your feedback and understand your concerns regarding the consistency of our results with published work. We carefully addressed your comments and revised the manuscript to enhance its scientific validity. Your input is invaluable, and we are committed to presenting accurate and reliable research.

Comment 1: - lines 105-110: a sentence is necessary to add which specifies that the distribution was hypothesized to be Gaussian based on past works (cite), and no confidence level with data was calculated,

Response: We sincerely appreciate your thoughtful comment. To address your valuable suggestion, we have revised the manuscript accordingly by adding a line, “Importantly, our decision to adopt a Gaussian distribution is grounded in prior research [13, 19]. The absence of a specific confidence level calculation can be attributed to our reliance on the Gaussian distribution assumption”

Comment 2: - line 259 is wrong because the Dst reached -60 nT on September 2, 2023, it should be modified;

Response: We have added a line making it clearer, “However, it should be noted that the Dst value reached -60 nT on September 2, 2023, which may suggest a moderate to relatively mild geomagnetic storm. Dst values less than -100 nT signify a severe or major geomagnetic storm. (NOAA).”

Comment 3: - the sentence on lines 262-264 does not agree with current research, a kp = 4 is able to influence TEC, see for example Liu H, Otsuka Y, Hozumi K and Yu T (2023) Day-to-day variability of the equatorial ionosphere in Asian sector during August–October 2019. Front. Astron. Space Sci. 10:1198739. doi: 10.3389/fspas.2023.1198739

Response: As per NOAA, the Kp index is a scale that ranges from 0 to 9, with Kp 5 indicating a transition from minor storm conditions to active geomagnetic storming. In earthquake studies, researchers often consider a range of Kp values. There is no universally agreed-upon threshold for what Kp level is suitable for such investigations, as the choice of a Kp threshold depends on the hypothesis being tested and the geographic region under study.

Researchers may seek patterns or anomalies in Kp values leading up to seismic events, but the significance of specific Kp levels can be highly context-dependent. For example, in the most recent study by Kotulak et al. (2021), the authors assessed GNSS ROTI (Rate of TEC Index) performance during solar cycle 24 geomagnetic disturbed periods. They utilized data from the Canadian High Arctic Ionospheric Network (CHAIN) and the Canadian Advanced Digital Ionosonde (CADI) to study ROTI performance, along with the Kp index to characterize geomagnetic activity during the observation period considering it below 5.

Kotulak, K.; Krankowski, A.; FroÅ„, A.; Flisek, P.; Wang, N.; Li, Z.; BÅ‚aszkiewicz, L. "Sub-Auroral and Mid-Latitude GNSS ROTI Performance during Solar Cycle 24 Geomagnetic Disturbed Periods: Towards Storm’s Early Sensing." Sensors 2021, 21, 4325.

In several common references related to earthquake studies, it is noted that they have set a Kp threshold of 5. In these studies, it is mentioned that the Kp index remained below 5 during the observation period, signifying a period of notably calm geomagnetic conditions. In our specific research context, the positive anomaly on September 2nd occurred at 13:7 UTC, coinciding with a time when the Kp remained consistently below 5.

Sharma, G., Champati ray, P. K., Mohanty, S., & Kannaujiya, S. (2017). "Ionospheric TEC modelling for earthquakes precursors from GNSS data." Quaternary International, 462, 65–74.

Sharma, G., Saikia, P., Walia, D., Banerjee, P., & Raju, P. L. N. (2020). "TEC anomalies assessment for earthquakes precursors in North-Eastern India and adjoining region using GPS data acquired during 2012–2018." Quaternary International.

Comment 4: the discussion should be rewritten accordingly;

Response: Modified accordingly

Comment 5: - Figure 2, could use only even or odd days even lasting all rods;

Response: We have modified the figure as asked with only even days

Comment 6: - Figure 4, rods and vertical shaded daily lines are necessary;

Response: we have input as suggested.

Comment 7: - Conclusions should be rewritten after considering that kp < 5 is not a sufficient condition to exclude the geomagnetic influence on TEC.

Response: We modified the conclusion section as advised, “It is crucial to acknowledge that, while we meticulously evaluated space weather conditions using specific indices (Dst, Kp, and F10.7) to focus exclusively on periods of relative calm, we recognize that Kp < 5 alone is not a sufficient condition to exclude the geomagnetic influence on TEC.”

Reviewer 2 Report

All my comments have been properly treated. No further comment. The manuscript is now ready for publication.

Author Response

Thank you for your feedback. I'm glad to hear that all your comments have been properly addressed, and the manuscript is now ready for publication. We highly appreciate your participation